# High-resolution TADs reveal DNA sequences underlying genome organization in flies

Fidel Ramírez [1], Vivek Bhardwaj [1,2], Laura Arrigoni [1], Kin Chung Lam[1], Björn A. Grüning [3], José Villaveces[4], Bianca Habermann[4], Asifa Akhtar[1] & Thomas Manke [1]

Despite an abundance of new studies about topologically associating domains (TADs), the role of genetic information in TAD formation is still not fully understood. Here we use our software, HiCExplorer (hicexplorer.readthedocs.io) to annotate >2800 high-resolution (570 bp) TAD boundaries in *Drosophila melanogaster*. We identify eight DNA motifs enriched at boundaries, including a motif bound by the M1BP protein, and two new boundary motifs. In contrast to mammals, the CTCF motif is only enriched on a small fraction of boundaries flanking inactive chromatin while most active boundaries contain the motifs bound by the M1BP or Beaf-32 proteins. We demonstrate that boundaries can be accurately predicted using only the motif sequences at open chromatin sites. We propose that DNA sequence guides the genome architecture by allocation of boundary proteins in the genome. Finally, we present an interactive online database to access and explore the spatial organization of fly, mouse and human genomes, available at http://chorogenome.ie-freiburg.mpg.de.

[1] Max Planck Institute of Immunobiology and Epigenetics, Stübeweg 51, 79108 Freiburg, Germany. [2] Faculty of Biology, University of Freiburg, Schänzlestraße 1, 79104 Freiburg, Germany. [3] University of Freiburg, Department of Computer Science, Georges-Köhler-Allee 106, 79110 Freiburg, Germany. [4] Max Planck Institute of Biochemistry and Computational Biology, Am Klopferspitz 18, 82152 Martinsried, Germany. Fidel Ramírez and Vivek Bhardwaj contributed equally to this work. Correspondence and requests for materials should be addressed to T.M. (email: manke@ie-freiburg.mpg.de)

How the DNA packs into the nucleus and coordinates functional activities is a long-standing question in biology. Recent studies have shown that the genome of different organisms is partitioned into chromatin domains, usually called topologically associated domains (TADs), which are invariable between cell types and evolutionary conserved in related species[1].

To understand TAD formation, researchers had focused on the proteins found at TAD boundaries[2–4]. In mammalian cells, the CCCTC-binding factor (CTCF) protein has been shown to be enriched at chromatin loops, which also demarcate a subset of TAD boundaries (referred to as "loop domains")[5]. A proposed mechanism, based on the extrusion of DNA by cohesin, suggests that the DNA-binding motif of CTCF and its orientation determine the start and end of the loop[6,7]. In line with this hypothesis, deletions of the CTCF DNA-motif effectively removed or altered the loop[6] or caused changes in gene–enhancer interactions that lead to developmental abnormalities in mouse embryos[8]. Additionally, acute depletion of CTCF leads to loss-of-TAD structure on CTCF containing boundaries[9]. However, CTCF-cohesin loops only explain a fraction (<39%) of human TAD boundaries[5], while plants and bacteria lack CTCF homologs but also show TAD-like structures. Thus, it is possible that additional factors are involved in the formation of TADs.

In contrast to mammals, the genetic manipulation tools available in flies have allowed the characterization of several proteins that, like CTCF, are capable of inhibiting enhancer-promoter interactions. Throughout the manuscript, we will refer to these proteins as "insulator proteins" and their binding motifs as "insulators" or "insulator motifs". In flies, apart from CTCF, the following DNA-binding insulator proteins have been associated to boundaries[3,10]: Boundary Element Associated Factor-32 (Beaf-32), Suppressor of Hairy-wing (Su(Hw)), and GAGA factor (GAF). Also, Zest white 5 (Zw5) has been proposed to bind boundaries[11]. These insulator proteins recruit co-factors critical for their function, such as Centrosomal Protein-190 (CP190) and Mod(mdg4)[12]. Recently, novel insulator proteins have been described as binding partners of CP190: the zinc finger protein interacting with CP190 (ZIPIC), Pita[13] which appear to have human homologs and localizes to TAD boundaries[11], and the Insulator binding factors 1 and 2 (Ibf1 and Ibf2)[14]. Except for CP190 and Mod(mdg4), all previously characterized boundary associated proteins bind to specific DNA motifs, suggesting that the 3D conformation of chromatin can be encoded by these motifs.

In this study, we sought to identify the DNA encoding behind TAD boundaries in flies. First, we develop software (HiCExplorer) to obtain boundary positions at 0.5 kilobase resolution based on published Hi-C sequencing data from *Drosophila melanogaster* Kc167 cell line[15,16]. Using these high-resolution TAD boundaries, we identify eight significantly enriched DNA-motifs. Five of these motifs are known to be bound by the insulator proteins: Beaf-32, CTCF, the heterodimer Ibf1 and Ibf2, Su(Hw) and ZIPIC. We find that a large fraction of boundaries contain the motif bound by the motif-1 binding protein (M1BP)[17], a protein associated to constitutively expressed genes. This motif has recently been found at boundaries[18,19]. The two remaining DNA-motifs have not been associated to boundaries before. Surprisingly, we find that depletion of Beaf-32 has no major effect on chromosome organisation, while the depletion of M1BP leads to cell arrest in M-phase and dramatically affects the Hi-C results. Using machine learning methods based on the acquired DNA-motif information, we could accurately distinguish boundaries from non-boundaries and identify TAD boundaries that were missed when using only Hi-C data. Our results suggest that the genome architecture of flies can be explained predominantly by the genetic information. We have implemented the methods for Hi-C data processing, TAD calling and visualization into an easy to use tool called HiCExplorer (hicexplorer.readthedocs.io). To facilitate exploration of available Hi-C data, we also provide an interactive online database containing processed high-resolution Hi-C data sets from fly, mouse and human genome, available at http://chorogenome.ie-freiburg.mpg.de.

## Results

**High-resolution TAD boundaries in flies**. We obtained Hi-C data for Kc167 cells from two recent studies[15,16] and processed them to obtain corrected Hi-C contact matrices at restriction fragment resolution (Methods section). These data sets contain the most detailed contact maps in flies, compared to other data sets (Methods section), due to high-sequencing depth (over 246 million valid read pairs) and the use of DpnII, a restriction enzyme with short-restriction fragment size (mean ~570 bp). We found 2852 TADs having a median size of 26 kb (Fig. 1a, Supplementary Figure 1a). We corroborated the precision of our boundaries by comparing their overlap with CP190 peaks (p-value = 1.8E − 20, Fisher's exact test, Fig. 1b) and the separation of histone marks (Fig. 1c, Supplementary Figure 2b). We classified TADs (Supplementary Data 1) using modENCODE histone marks[3] as active (enriched for either H3K36me3, H3K4me3, and H4K16ac), polycomb group silenced (PcG) (enriched for H3K27me3), HP1 (enriched for H3K9me3) and inactive (not enriched for any of the marks, see Methods section and Supplementary Figure 1c). A significant fraction of the genome (43%) is covered by large inactive TADs having a mean length of 63 kb (Fig. 1d). In contrast, active chromatin TADs have a mean length of 23 kb and occupy 29% of the genome. PcG chromatin occupies 25% of the genome with TADs that are on average 61 kb. The largest TADs are found for HP1 repressed chromatin, which together occupy 3% of the genome and have a mean length of 74 kb. We also find that active TADs tend to be assembled one after the other due to their higher number (Fig. 1e). Interestingly, the TAD separation score varied significantly (p-value <7.8E − 5, Wilcoxon rank-sum test) between the TAD types (Fig. 1f). The stronger boundaries (low-TAD-separation score) are found between active and inactive or PcG TADs. While the weakest boundaries are found between PcG TADs. Similarly, we find that the TAD-separation score between larger TADs (mostly inactive) is significantly larger than the TAD separation score for smaller TADs (mostly active) (p-value = 9.9E − 7, Wilcoxon rank-sum test).

While most of our boundaries overlap with those from previous studies[3,16], our method allowed us to identify a larger set of boundaries (Supplementary Figure 1d, e) with increased precision (as measured by distance to CP190, Supplementary Figure 1f–h), which mostly subdivide active TADs from previous studies. We observed that the majority of the boundaries (77%) are located at gene promoters (henceforth referred to as promoter-boundaries. Figure 1g). Promoter-boundaries are different from non-promoter boundaries (23%), since they associate significantly with active chromatin (Fig. 1h, H3K4me3 p-value = 4.43E − 144 Wilcoxon rank-sum test, other histone marks can be seen in Supplementary Figure 1i), have lower TAD separation score representing stronger boundaries (p-value = 8.5E − 35, Wilcoxon rank-sum test, Fig. 1i), and show higher DNAse sensitivity (Fig. 1j).

**Specific gene orientation and transcription marks boundaries**. Next we correlated our promoter boundaries with gene orientation and transcription. Most promoter-boundaries (70%, p-value = 2.5E − 88 Fisher's exact test), are marked by divergently

oriented gene promoters on either side, while genes in convergent or tandem orientation tend to be inside the TADs. To correlate TADs with gene transcription, we analysed the RNA-Seq data from 14 stages of *Drosophila* development along with the expression in the Kc167 cell line obtained from modENCODE

(Supplementary Figure 2a, Methods section). We found that 95.6% of genes which have a TAD boundary at their promoter are expressed in Kc167 cells (1244 out of 1300) compared to 75.3% of genes which do not have a boundary at their promoter (6892 out of 9149, *p*-value = 2.19E − 80, Fisher's exact test). Higher

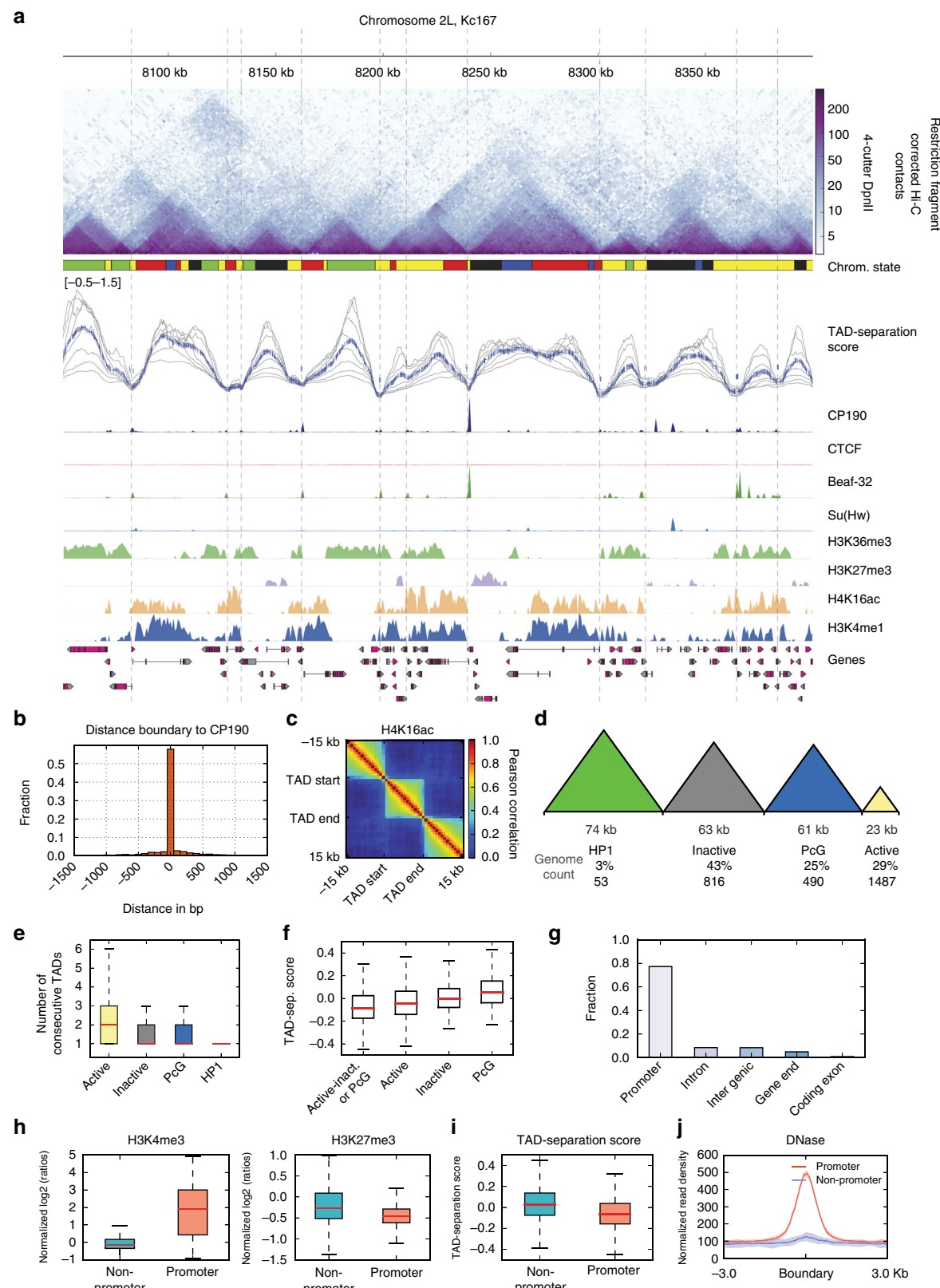

correlation was observed between gene expression inside TADs than between neighboring TADs (Fig. 2a, Supplementary Figure 2c). When we investigated the expression of genes at TAD boundaries we found that these genes show significantly higher expression than the expressed genes which do not have a TAD boundary at their promoters (Fig. 2b, p-value = 6.08E − 21, t-test). Boundary associated genes also show a more stable expression across development than other genes (Fig. 2c, Supplementary Figure 2b), suggesting that these genes are ubiquitously transcribed. Furthermore, we find that for a pair of genes lying next to each other, the variability in their gene expression tend to be correlated during development if these genes are within same TAD, while this correlation is lost if there is a TAD boundary in between (Fig. 2d, Methods section). This is true for gene-pairs in any orientation (convergent, divergent, or tandem, Supplementary Figure 2d).

Taken together, these results suggest that specific gene orientation and level of transcription could be associated with TAD formation.

**A comprehensive list of boundary associated DNA motifs.** We followed the strategy outlined in (Fig. 3a) to create a comprehensive list of motifs frequently found at boundaries. First, we performed de novo motif calling using MEME-chip[20] (Methods section) on our promoter-boundaries and non-promoter boundaries. To filter out motifs that are frequently found at promoters or open chromatin but are not specific to boundaries, we performed an enrichment analysis using two different methods: Ame[21] and TRAP[22]. As a second approach, we tested boundaries for the motifs of known insulator proteins and core-promoter motifs from a previous study[23]. In contrast to de novo motif detection, searching for known motifs allows additional sensitivity to detect low frequency motifs. After filtering for only consistent results, we could identify 5 motifs enriched at promoter-boundaries and 3 motifs enriched at non-promoter boundaries (Fig. 3b).

The promoter boundary motifs we identified belong to the list of core-promoter motifs 1, 2, 6, 7, and 8 from Ohler et al.[23]. Motif-1 is recognized by the recently described 'motif-1 binding protein' (M1BP), a protein found at the promoters of transiently paused Pol-II of constitutively expressed genes[17]. Motif-2 (also called DRE motif) is recognized by the insulator protein Beaf-32 and DREF[24]. Motif-7 (also called DMv3) is recognized by the insulator protein ZIPIC. The binding proteins for motif-6 (also known as DMv5) and motif-8 (also known as DMv2) are, to the best of our knowledge, not known. De novo motif calling also identified other core-promoter motifs but they were not found enriched at boundaries. The three motifs that we find enriched on non-promoter boundaries, correspond to the binding sites of Su

(Hw), CTCF and Ibf (Fig. 3b). We could not find an enrichment for motifs of other insulator proteins like GAF, Pita, and Zw5. For clarity, we will refer to the boundary motifs by the name of the insulator protein that binds to them, except for motif-6 and 8.

As an independent validation we find that our three predicted boundary motifs (M1BP motif, motif-6, and motif-8) are also enriched at the binding sites of CP190 and Cap-H2 (condensin II complex) (Supplementary Table 1). We repeated our analysis using the TAD boundaries from previous studies[3,15,16] and found similar enrichments (Supplementary Table 2).

To better understand the distribution of the motifs on boundaries we performed hierarchical clustering of the binding affinity (TRAP score) for the eight motifs enriched at boundaries (Fig. 3c left panel, Supplementary Figure 3a). We then plotted the ChIP-seq signal of the DNA-binding proteins (Fig. 3c second panel, Supplementary Figure 3b), along with CP190, Cap-H2, Rad21 (Fig. 3c third panel) and RNA Pol-II, over the clusters. The results show that the boundary motifs are usually associated with their corresponding proteins, except for motifs 6 and 8 for which the binding proteins are not known (Fig. 3c second panel). ZIPIC and Ibf show ChIP-seq enrichment at many regions that do not have the motif although their enrichment is higher when the motif is present. This could indicate indirect binding of the proteins as seen for CP190 or could also indicate antibody cross-reactivity or other problems with ChIP-seq experiments[25,26]. Examples of the boundaries with their motifs and corresponding proteins can be seen in Supplementary Figure 3c–g.

We discover that promoter boundaries are primarily associated with Beaf-32 and M1BP motifs. Promoter boundaries also tend to be associated with condensin II (Cap-H2), RNA polymerase II and housekeeping enhancers. Interestingly, Rad21 (cohesin) ChIP-seq peaks mostly associate with M1BP motif (Fig. 3c–d) and de novo motif calling on Rad21 peaks identified a clear enrichment for M1BP motif (MEME[27] E-value = 4.2E − 97) (Supplementary Table 1). Furthermore, M1BP ChIP-seq signal correlates well with Rad21 ChIP-seq (Supplementary Figure 3h). Thus, while Cap-H2 is found together with all boundary promoter motifs, Rad21 is closely associated with M1BP. We found that specific boundaries containing M1BP, as well as ZIPIC and Motif-6 are also associated with paused Pol-II (Fig. 3e) (previously, only M1BP was associated to paused Pol-II[17]).

Ibf1, CTCF and Su(Hw) are the most common insulator motifs found at non-promoter boundaries, and tend to be associated with enhanced binding of CP190 co-factor (Fig. 3c–d). The ChIP-Seq signals for these proteins follows this observation, showing that promoter and non-promoter boundary proteins are correlated within their group (Supplementary Figure 3h).

We then searched for association of all the transcription factors from modENCODE consortium[28], as well as from the

**Fig. 1** High-resolution TAD boundaries in flies. **a** Example region of 350 kb showing Hi-C TADs from Kc167 cells. Top panel: Hi-C contact matrix obtained from refs. [15,16]. The size of the bins is variable (mean 570 bp) and depends on the genomic location of the DpnII restriction sites. The chromatin state track contains the five classifications from ref. [29]: Active chromatin, red and yellow; inactive chromatin, black; PcG, blue; HP1, green. The TAD separation score track (Methods section) depicts a normalized measure of the contacts between two flanking regions (10–40 kb, depicted by gray line, blue line depict mean score). The boundaries, estimated using the TAD separation score are shown as vertical lines. The following tracks show normalized ChIP-seq coverage for the known boundary proteins CP190, Beaf-32, and Su(Hw) on Kc167[39] and CTCF[15]. The following tracks contain ChIP-chip data for histone modifications from modEncode[28]. The image was generated using HiCExplorer. This particular region was selected because many different TADs could be seen; other regions can be browsed at http://chorogenome.ie-freiburg.mpg.de. **b** Histogram of the distance of a boundary to the nearest CP190 (common insulator protein co-factor) peak. **c** Correlation of histone marks within and between TADs. Each pixel in the matrix represent the Pearson correlation of the histone mark in all TADs at different distances (Methods section). **d** TAD classification based on histone marks. The numbers below each TAD type represent respectively: mean length, percentage of genome occupied by the TAD and number of TADs of that type. **e** Boxplot of consecutive TAD of each type. **f** TAD-separation score between: active and inactive or PcG, active–active, inactive–inactive, and PcG–PcG. The differences between the groups are all significant (p-value < = 7.8E − 5, Wilcoxon rank-sum test). **g** Classification of TAD boundaries. TAD boundaries are classified at promoter if they are within 1000 bp of the annotated TSS. **h** Histone marks at non-promoter and promoter boundaries. Further marks can be seen in Supplementary Figure 1i. **i** TAD-separation score for non-promoter and promoter boundaries. **j** DNase accessibility at non-promoters and promoter boundaries

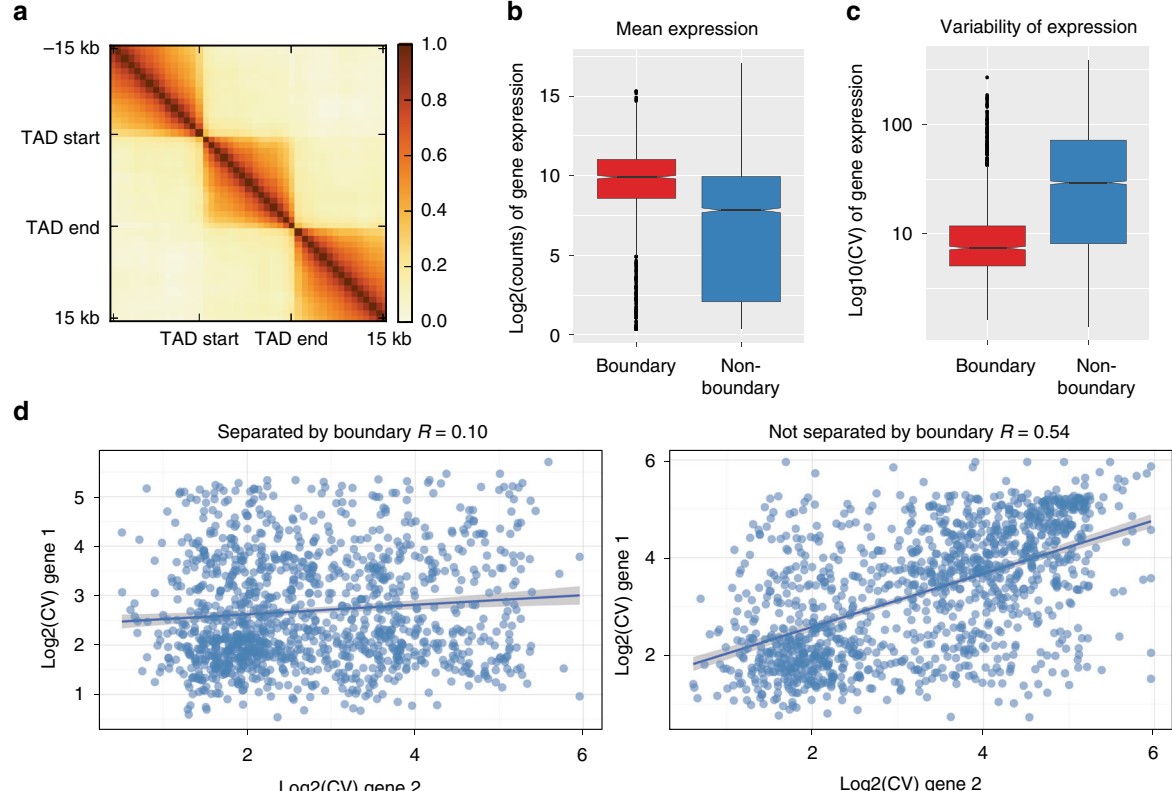

**Fig. 2** TAD boundaries are marked by specific gene orientation and transcription. **a** Correlation of mean expression across developmental stages inside TADs vs. outside. Region inside TADs was scaled to 15 kb. Each pixel in the matrix contains the pearson correlation at different distances. **b**–**c** Mean expression (in Kc167 cells) and variability of expression (during development) for genes whose promoters are at a TAD boundary vs. genes whose promoters are not at boundaries. **d** Coefficient of variation (across developmental stages) between pairs of adjacent genes either separated by a TAD boundary (left) or not separated by a boundary (right). Line shows the linear model fit (shaded region: std. error)

comprehensive collection of ChIP-seq data from refs. [15,16] with our boundary motifs. Interestingly, our screen for proteins associated to boundary motifs showed that Nup98, a component of the nuclear pore complex, is associated with motif-6 and Pita, along with CTCF (Fig. 3d and Supplementary Figure 3i). Further validation using de novo motif calling on Nup98 peaks identified motif 6 and Pita motif as the most enriched motifs (MEME[27] E-value = 3.4E−4 and 3.3E−9, respectively).

We observed that ChIP-seq peaks of DNA-binding proteins are often found in regions not containing their motif (Fig. 3d). For example, ZIPIC peaks can be seen together with motif 6 or CTCF although ZIPIC motif does not overlap with any of them (Fig. 4a). Similar observations can be made for CTCF ChIP-seq experiments (Supplementary Figure 3j, see discussion), suggesting that motif sequences should be considered along with ChIP-Seq binding sites as more reliable functional predictors of an insulator protein.

**Motif combinations reflect boundary strength**. Next we looked at motif combinations at boundaries in relation to boundary strength. Using ChIP-Seq analysis it has been reported that boundary strength increases with the number of proteins bound at the boundary[10]. When we looked for motif combinations at boundaries that overlap with their corresponding protein, we did not observe any significant differences in boundary strength between boundaries containing one, two, or three motifs (we only found eight boundaries having more than three motifs, Supplementary Figure 4b), although we could replicate earlier results based on ChIP-seq peaks (Supplementary Figure 4c). We found

that specific motifs or their combinations are associated with boundary strength (Fig. 4a). Boundaries containing the motif for Ibf, Su(Hw) or the combination of the two motifs are weaker than average while the combination of the Beaf-32 motif with either Pita, ZIPIC, or motif-6 result in the strongest boundaries.

We also looked at the association of motifs with active, inactive, PcG and HP1 TADs from Fig. 1d. We observe that the promoter-boundary motifs are mostly found between active TADs or between active and inactive (including PcG) TADs. Conversely, the non-promoter boundary motifs are rarely found between active TADs, and mostly separate active from inactive TADs or are between inactive TADs (Fig. 4b). We find the same trend when analysing the ChIP–chip log2 ratios of the active histone mark H3K36me3 and the repressive histone mark H3K27me3 surrounding the boundaries (Supplementary Figure 4d). Most promoter-boundary motifs separate active-inactive marks, while most non-promoter boundary motifs lie within inactive marks.

Additionally, we analyzed the chromatin state[29] that overlaps with the boundary motifs and found that promoter-boundary motifs lie mostly within active chromatin while non-promoter boundary motifs lie within both active and inactive chromatin (Fig. 4c).

**Boundaries can be predicted using motifs**. To better characterize boundaries at promoters we used three standard classification methods and ranked features by their relevance to distinguish boundaries from other promoters (Methods section). The features included the TRAP score of all motifs studied along

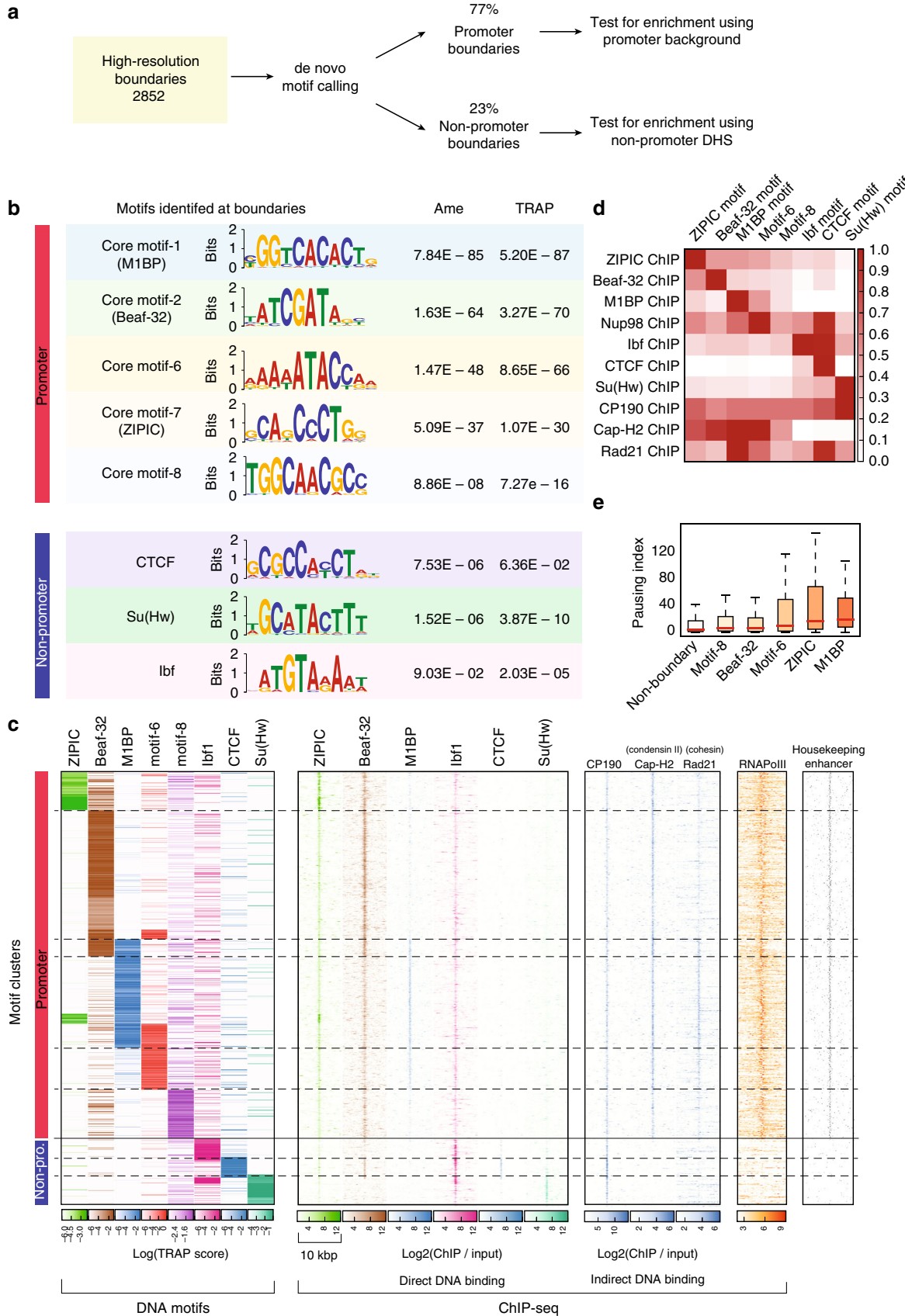

with other known insulator motifs. We also used DNase hyper-sensitive sites (DHS) as a feature to identify open promoters, considering that protein binding to a motif requires an accessible promoter. The ranking of feature importance showed that indeed open promoters are required for boundaries (Fig. 5a). For the motifs, the feature importance ranking follows the abundance of the motifs at boundaries: M1BP and Beaf-32 show the highest importance scores, followed by motif-6, ZIPIC, and motif-8. Some features, like GAF and Pita motifs were found to be negatively associated with boundaries at promoters (Supplementary Figure 5a).

Although regularized models used here are less prone to overfitting compared to other machine learning methods, we further protect against overfitting by 10-fold cross validation during training. We then tested model accuracy on an independent test data set. The classifiers showed a sensitivity and specificity over 71% on the test data (Fig. 5b). We also classified open chromatin regions distant from promoters, using only motif scores on these sites as features (Fig. 5c). This resulted in a sensitivity and specificity over 60% (Fig. 5d). The motif binding affinity can also be used as a linear predictor of boundary strength (Supplementary Figure 5b, d). Interestingly, we find the motif bound by GAF, a protein recognized to bind insulators[30,31], to be negatively associated with both promoter and non-promoter boundaries (Supplementary Figure 5c, e).

The machine learning predictions can complement the Hi-C derived boundaries at the promoters, as some boundaries predicted by our model are missed by our boundary detection method based on the TAD-separation score (Fig. 5e).

**Depletion of Beaf-32 protein does not affect TAD boundaries.** Our analysis showed that Beaf-32 and M1BP motifs are enriched at *Drosophila* boundaries. We further explored the effect of depletion of the proteins that bind these motifs on chromosome organisation by performing siRNA mediated depletion of Beaf-32, M1BP or both, followed by in situ Hi-C protocol (Methods section). We used the knockdown of GST as a negative control and two biological replicates for each condition. We first observed that while Beaf-32 depletion has no effect on M1BP protein levels, M1BP knockdown leads to downregulation of Beaf-32 (Supplementary Figure 6a). We therefore expected M1BP knockdown to produce either similar, or more exacerbated effects compared to those in Beaf-32 knockdown. Surprisingly, Hi-C analysis showed no global change in chromosome conformation upon knockdown of Beaf-32 compared to GST control (Supplementary Figure 6b). This effect was highly reproducible (Supplementary Figure 6c–f). Depletion of M1BP showed a dramatic effect on the distribution of Hi-C contacts in which short range contacts decrease and inter-chromosomal interactions increase (Supplementary Figure 6d, h). We further found that M1BP knockdown cells were arrested in M-phase (Supplementary Figure 6g), yet the contact distributions are different than those observed for fly mitotic cells

from Hug et. al.[18] (Supplementary Figure 6h). The fraction of inter-chromosomal contacts is often used as a quality measure for Hi-C data[32]. For healthy cells it had been suggested that this fraction should be <20% of all valid reads. This assumes that chromosomes occupy their own territories which may not be true for perturbed cells. Our data from wild-type cells and Beaf-32 knockdown show an average of 8% inter-chromosomal interactions. The samples from M1BP knockdown and double knock-down show a replicable high fraction of inter-chromosomal contacts (~45%, Supplementary Figure 6d). Since they were identically processed in the same batch and alongside the wild type and Beaf-32 samples, we argue that this is not a technical artifact and the large number of inter-chromosomal contacts probably reflects a biological effect on chromatin.

**Resources to explore TADs and associated genomic features.** During our research, we developed processing and analysis tools for chromosome conformation. Our tool-suite, called HiCExplorer, simplifies the Hi-C data pre-processing, quality controls, contact matrix transformation, and TAD calling into a few easy steps (Fig. 6a). HiCExplorer is open source and is available at https://github.com/deeptools/HiCExplorer/. Importantly, HiCExplorer can be used with other pipelines and processing tools as we have built-in import/export functions covering commonly used Hi-C data formats. To facilitate analysis, we have integrated HiCExplorer into the Galaxy platform[33]. With HiCExplorer, we made available our efforts to create meaningful and accurate visualizations of Hi-C data that can integrate other data sources, examples of which can be seen throughout this manuscript. Further information can be found at the associated documentation (http://hicexplorer.readthedocs.io), which includes a full analysis workflow and detailed description of the tools.

Our tools would be beneficial to users that do not routinely perform expensive and technically challenging Hi-C experiments. They enable quick visualization of specific genes or regions of interest in the context of TADs and loops, to understand gene regulation. We provide a resource called the Chorogenome Navigator (Fig. 6b) (http://chorogenome.ie-freiburg.mpg.de/), which includes *Drosophila*, Mouse, and Human Hi-C data sets, already processed by HiCExplorer, along with associated gene annotations, histone marks and other TAD/boundary annotations. The underlying program called HiCBrowser (https://github.com/deeptools/HiCBrowser/), is also freely available to be used as a standalone browser, where users can include their own genomic tracks. With these resources, we hope to make Hi-C analysis a routine part of genomics workflows.

## Discussion

In this study, we used high resolution (DpnII restriction enzyme) and deeply sequenced (~246 million reads) Hi-C data[15,16] to map the genomic positions of TAD boundaries within ~600 bp in *D.*

**Fig. 3** Eight motifs are enriched at boundaries. **a** Overview of the strategy used to identify de novo motifs. **b** Motifs enriched at promoter and non-promoter boundaries (along with Bonferroni-corrected *p*-values). Two methods were used to estimate enrichment (Methods section): Ame[21] and TRAP[22]. **c** Clustering of boundaries by motif binding affinity (Methods section). Each row represents one boundary. Left panel: clustering of motif binding affinity using the TRAP score[22]. Higher scores indicate stronger predicted binding. Dashed lines delineate the clusters. Following panels: using the motif clustering results, we show the heatmaps corresponding to ChIP-seq enrichments for insulator proteins binding the DNA (second panel), other proteins that bind indirectly (third panel) and RNA Pol-II. Last panel shows housekeeping enhancers from ref. [69]. For boundaries at promoters, heatmaps are centered at the gene promoter, for non-promoter boundaries, heatmaps are centered at the nearest CP190 peak within 2000 bp. ChIP-seq signal was computed in 50 bp bins for 5000 bp from the center. The scale of each heatmap goes from 1 to 12 for the direct DNA-binding ChIPs and from 1 to the max ChIP-seq value for the indirect binding ChIPs (based on Supplementary Figure 3b). **d** Relationship between motif presence and ChIP-seq peak fold change at boundaries. Each cell in the matrix contains the mean fold change of all respective ChIP-seq peaks having the motif. For each row, the maximum fold change was scaled to 1. **e** Pausing index at different boundary-promoters containing one of the boundary motifs. Non-boundary promoters are plotted as control

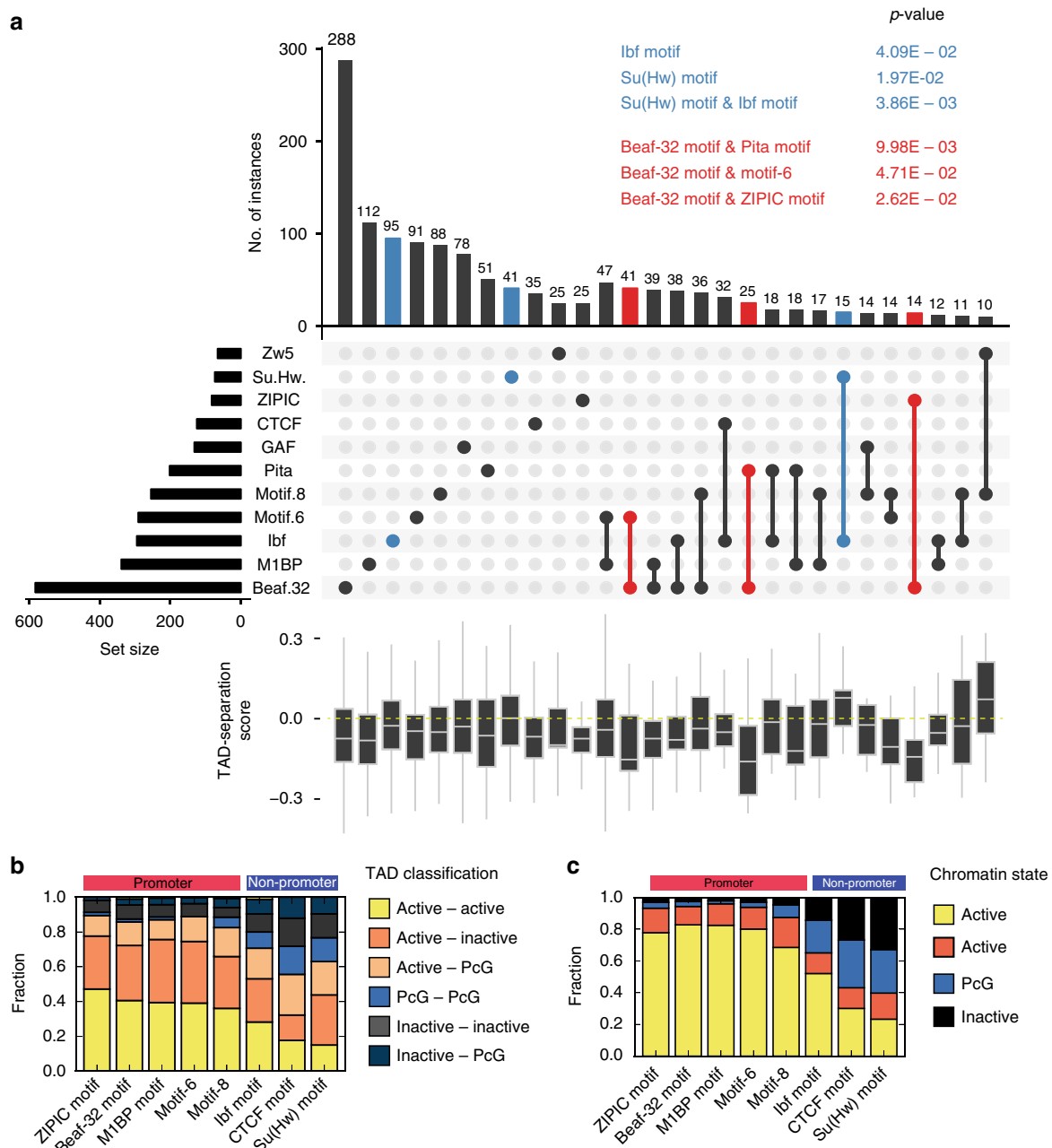

**Fig. 4** Promoter and non-promoter boundary motifs show marked differences. **a** TAD-separation score at boundaries grouped by the motif presence. For this analysis we considered a motif to be present if the motif overlaps with a ChIP-seq peak (Methods section). The bars show the overlap between the indicated motifs below. The boxplots show the distribution of the respective TAD-separation score. The sets highlighted in blue have a TAD-separation score distribution significantly larger than the overall TAD-separation score. The p-values (Wilcoxon rank-sum test) are shown above the figure. Similarly, the sets highlighted in red have a distribution significantly smaller. Only motif combinations having >10 instances are shown. Motif combinations with three or more motifs were rare. The intersections were plotted using UpSetR[70]. An overview of the motif overlaps can be seen in Supplementary Figure 4a. **b** Frequency of flanking TAD types (as classified in Fig. 1d) per boundary motif. **c** Frequency of the chromatin state from ref. [29]

*melanogaster*. Our analysis revealed a larger number of TADs, including many small active TADs (23 kb mean length), that were absent in previous reports[3,4,34]. We characterized TAD size, boundary strength, chromatin marks, gene orientation, and transcription at the TADs. We perform motif calling at boundaries, validating the presence of known insulators, along with M1BP motif, which recently has also been shown to be associated to boundaries[18,19] and core promoter motif 6 and motif 8, which have not been associated to boundaries before. Using different machine learning methods, we find that DNA motifs and open chromatin are sufficient to accurately predict a major fraction of fly boundaries. Finally, we present a set of useful tools and a resource for visualization and annotation of TADs in different organisms.

Our study verifies various properties of fly boundaries indicated in previous publications. We detect that most boundaries associate with promoters and active chromatin (Fig. 1g, h)[4] and that various known insulator proteins are enriched at boundaries (Fig. 3b,c)[3,4]. We also detect a comprehensive set of core promoter motifs at boundaries, including the newly discovered

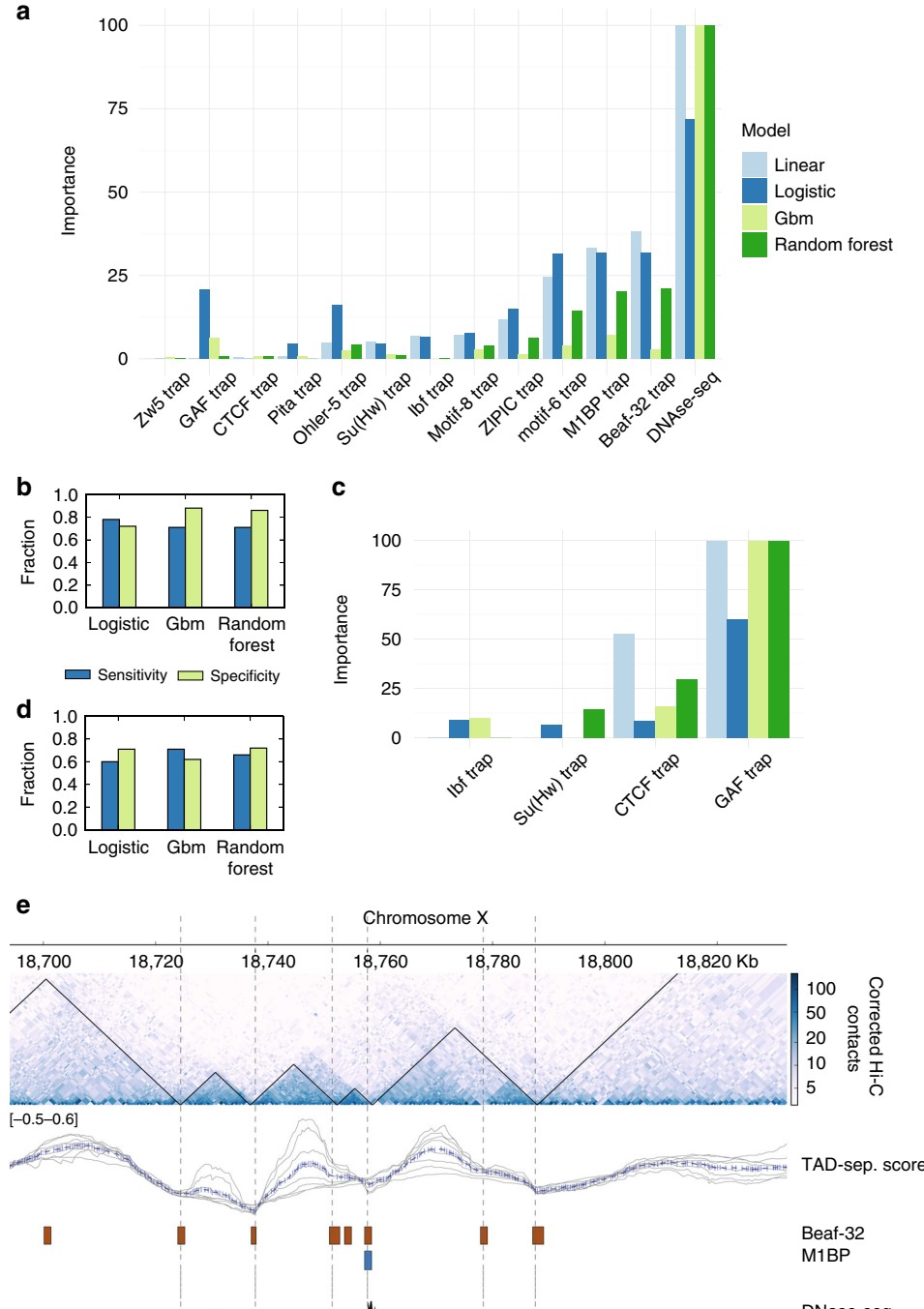

**Fig. 5** Prediction of boundaries using machine learning. **a** Feature importance for promoter boundaries computed using four different methods: linear model, logistic regression, gradient boost model (gbm), and random forest. Importance scores of each method were scaled from 0 to 100. Except for DNAse-seq, each feature represents the binding affinity (TRAP score) of the respective motif. **b** Sensitivity and specificity for promoter boundaries measured for logistic regression, gradient boost model and random forest. The output of the linear model can be seen in Supplementary Figure 5b. **c** Feature rankings, as in **a**, for non-promoter boundaries. **d** Sensitivity and specificity for non-promoter boundaries. **e** Examples of high-resolution boundaries and predicted boundaries. The high-resolution boundaries (based on the TAD-separation score) are depicted as black triangles on top of the Hi-C heatmap. The predicted boundaries are shown as dotted vertical lines. The tracks below the Hi-C contact map contain the instances of the motifs that overlap with promoters. To aid the visualization of the short motifs, their genomic location was extended by 500 bp in each direction. The last track depicts regions of open chromatin based on DNase-seq from modEncode[28]

M1BP motif[17]–[19], and motifs which have been associated to housekeeping gene expression[35,36]. However, some of our results contradict previous observations. For example, we find that genes at boundaries have higher expression and lower variability of expression throughout fly development (Fig. 2b, c, Supplementary Figure 2b). This in line with Ulianov et al.[18,34] and Hug et al.[18,34] but in contrast with Hou et al.[4], who suggest that gene density and not the transcriptional state is important for boundary formation. Unlike Hou et al.[4] we find that genes at boundaries tend to be divergently transcribed. In contrast to various earlier

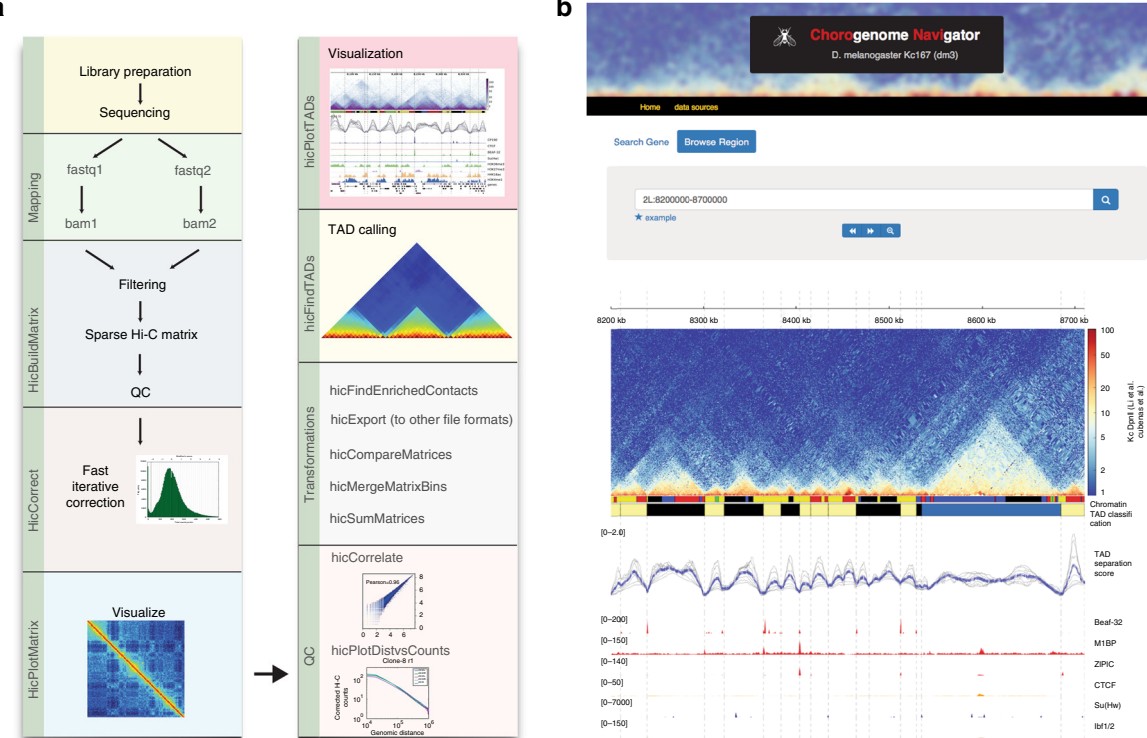

**Fig. 6** HiCExplorer and Chorogenome navigator. **a** Pipeline of Hi-C processing and the commands used by HiCExplorer. HiCExplorer is easy to install (*conda install hicexplorer*), easy to use, tested (automatic tests and numerous data sets processed), well documented (https://hicexplorer.readthedocs.io/) and is ready to be used within the Galaxy framework[33]. **b** The Chorogenome navigator aims to be a repository for available Hi-C data. Currently, we host data from fly, mouse and human. Customized tracks can be created using the HiCBrowser (https://github.com/deeptools/HiCBrowser/)

studies[3,4,37], CTCF does not appear to be a major boundary associated insulator in flies (Figs. 3c, 4a). We also show that the number of insulator motifs at boundaries correlates very little with boundary strength (Supplementary Figure 4b), in contrast to Van Bortle et al.[10].

Most of these differences are due to the increased resolution of detected boundaries (Fig. 1b, Supplementary Figure 1f–h) and the combined analysis of DNA motifs with ChIP-seq data, rather than ChIP-Seq peaks alone. We show that correlating boundaries with ChIP-Seq peaks alone is not a good measure when it comes to determinants of boundary formation. Many DNA-binding proteins show co-localization in ChIP-Seq data without presence of the corresponding DNA motifs (Fig. 3d, Supplementary Figure 3j). This is possible due to cross-linking artifacts and indirect binding, which is, in fact, aggravated at boundaries, which tend to contact each other in 3D space[38].

Another argument for considering motifs is the contradicting case of CTCF at boundaries. In contrast to earlier studies based on CTCF ChIP-seq[3,4], we find that the CTCF motif is rarely associated to boundaries. This difference is caused by the quality of the ChIP-seq data that can produce spurious peaks. For example, we could observe a significant enrichment of CTCF at boundaries in the ChIP-Seq data from Wood et al.[39] but not in the CTCF ChIP-seq data from Li et al.[15]. On the other hand, both ChIP-seq data sets show significant enrichment if we only consider ChIP-seq peaks that contain the CTCF motif, (Supplementary Figure 3j). For CTCF, and in general for ChIP-seq experiments in flies, 'phantom peaks' are known to occur at active promoters[25]. Thus, to avoid misleading results our analyses are based on motif presence when possible and for ChIP-Seq data sets, we use significance threshold along with motif binding affinity for analysis (instead of taking a significance cutoff alone).

We observe that boundary strength is associated with the chromatin states of flanking TADs and particular motif combinations, but is not associated with the number of co-occurring boundary motifs. Boundary strength is higher between active and inactive/PcG TADs while is lower at boundaries separating two TADs within the same state (e.g., active–active, inactive–inactive, Fig. 1f). Boundaries containing Beaf-32 are stronger when present together with either motif 6, Pita, or ZIPIC motif while weaker with motif 8 (Fig. 4a). Although, the mechanism by which combinations of insulators alter the boundary strength still remains unclear, we observe an association of Nup98 with Pita motif, motif 6, and CTCF (Fig. 3d and Supplementary Figure 3i), suggesting that association with nuclear pore proteins may result in stronger boundaries. Nup98 has now been shown to be functionally important in mediating enhancer-promoter looping in the *Drosophila* genome[40].

Our results indicate that the two sets of boundary motifs (promoter and non-promoter) participate in the compartmentalization of different types of chromatin. Boundaries containing core promoter motifs are either flanking, or surrounded by active chromatin regions (Fig. 4b). In contrast, the boundaries containing non-promoter motifs tend to be within or at the borders of inactive or repressed chromatin (Fig. 4b). This finding is in line with previous reports showing an enrichment of CTCF at the borders of H3K27me3 domains[3,37] and an enrichment of Beaf-32 in active chromatin[3]. This indicates that insulator proteins might serve different functions guided by the DNA sequence. For example, we observe that GAF motif, whose presence is negatively associated with TAD boundaries (Supplementary Figure 5a, c), is rather detected alone at "loop domains"[5] (Supplementary Figure 5e).

Our analyses indicate that the depletion of the well-studied insulator protein Beaf-32, has no significant effect on the chromosome conformation. However, in *Drosophila melanogaster*, both the Beaf-32 and DREF proteins bind exactly the same DNA motif[24]. Thus, our results, as well as others[41–43] point out that DREF, a protein that unlike Beaf-32 is conserved in humans, might have a more prominent role in genome organization than previously thought.

On the other hand, cells under M1BP knockdown grow slower in culture and get arrested in M-Phase, probably because M1BP is a transcription factor of constitutively expressed genes[17]. Since M1BP depleted cells show cell cycle defects, it is difficult to separate the direct role of M1BP at boundaries from the indirect effects caused by deregulation of thousands of genes. To study the direct role of M1BP at boundaries, it would be useful to perform either deletion of M1BP motif on boundaries using CRISPR, as shown for CTCF in mammals[44,45] or through acute and complete depletion of M1BP[9].

In this study, we present evidence that the DNA sequence contains features that can guide the formation of higher order chromosome organisation. The association of boundary types with a combination of motifs (Fig. 4a), and the fact that we can predict boundaries using DNA sequence alone, in absence of any information about associated protein or histone marks (Fig. 5 and Supplementary Figure 5a–d), leads us to propose a DNA-guided chromatin assembly model. In this model, the boundary elements are recognized by their proteins, which help loading TAD assembly factors onto chromatin. Promoter and non-promoter boundaries can thus have different mechanisms of formation. DNA motifs at inactive regions can attract proteins that may establish TAD domains by setting up barriers for chromatin marks[46]. Although overall barrier activity of insulator proteins have been controversial[10,43], it is plausible that the barrier mechanism is active only at a subset of boundaries (like those of inactive TAD domains). DNA motifs at gene promoters can associate with core-promoter proteins which then guide the assembly of Pol-II pre-initiation complex. The pre-initiation complex can then recruit condensins[47]. Once recruited, condensins can perform loop extrusion independent of Pol-II transcriptional activity[18,48,49], leading to emergence of TADs[6,50]. Condensins can also remain associated to chromatin during mitosis, to re-establish TADs after the cell division. In general, our results indicate that active transcription and chromosome conformation are related, in-line with a recent study[51]. Future studies investigating the association of Pol-II pre-initiation complex and condensin activity on gene promoters would advance our understanding of mechanism of TAD formation.

## Methods

**Hi-C processing**. Different Hi-C data available for *D. melanogaster* were downloaded from GEO and processed using HiCExplorer (http://hicexplorer.readthedocs.io/). The list of data sources can be seen in Supplementary Table 3.

Each mate of the Hi-C sequencing read pairs was aligned separately using bwa mem with parameters '-E50 -L0'. The E parameter is the gap extension penalty, which is set high to avoid gapped alignments. This is because a fraction of the reads from a Hi-C experiment contain sequences from two distinct genomic positions. By increasing the gap extension penalty we promote the aligner to map the two parts of the read separately instead of trying to map the read to a single location. The L parameter is the penalty for 5′ and 3′ clipping which we set to zero to favor such clipping for the same reason as before.

To create the contact matrices, HiCExplorer divides the genome into bins of unequal size demarcated by the genomic positions of the restriction site and a matrix was created having these bins as rows and columns. The mapped reads were processed to count the number of times any two bins were connected by the Hi-C reads pairs. The following reads were discarded: read pairs that were not uniquely mapped or had a mapping score lower than 15, were within 800 bp to each other, were duplicated, contained a dangling end indicative of defective re-ligation or when one of the fragment mates was farther than 1500 bp from the restriction site.

In our processing of the data, we observed that restriction enzymes do not cut with the same efficiency at all sites or sometimes do not cut at all. Because of this, after the creation of the contact matrices, rows, and columns with zero or small total counts were removed. To define a sample-specific lower cutoff, we analyzed the bimodal distribution of total counts per rows, which is a convolution of two distinct distributions. The distribution with lower counts contains all bins with zero reads, mostly from repetitive regions, and also bins with low number of reads, probably from inefficient digestion of restriction sites. We chose as count threshold the position of the local minimum between the two modes. After filtering low count bins, the matrices were corrected following the iterative procedure from ref. [52].

For the 4-cutter DpnII restriction enzyme the mean fragment length after removing low coverage bins is 570 bp. For the 6-cutter HindIII the average was 4500 bp.

**Identification of boundaries**. TAD boundaries were identified using an improved version of TAD-separation score method from ref. [53] which is similar to Top-Dom[54]. The method works by first transforming the Hi-C contact matrix into a z-score matrix $\mathbf{A} = (a_{ij})$. For this, each contact frequency in the matrix is transformed into a z-score based on the distribution of all contacts at the same genomic distance. For a bin $l$, the contacts between an upstream and downstream region of length $w$ are in the z-score submatrix of $\mathbf{A}[\alpha_l, \beta_l]$, such that $\alpha_l \in \{l - w, ..., l\}$ and $\beta_l \in \{l, ..., l + w\}$. This submatrix corresponds to the 'diamond' seen in Supplementary Fig. 1a. For each matrix bin we compute the TAD-separation score($w$) as the mean overall $\mathbf{A}[\alpha_l, \beta_l]$ values.

To reduce noise, a multi-scale version of the TAD-separation score is computed for different values of $w$ that are averaged per bin, $w \in \{10,000, 12,000, 18,000, 25,000,$ and $40,000\}$. Genomic bins with a low-TAD-separation score with respect to neighboring regions (local minima) are indicative of TAD boundaries (Supplementary Figure 1a) and stronger boundaries will have lower scores. To assign a statistical significance to each local minimum we compare, the distribution of the z-scores for the submatrices $\mathbf{A}[\alpha_l, \beta_l]$ having $l \in \{i, i - v, i + v\}$, where $i$ is the bin of the local minima, and $i - v$ and $i + v$ are the bins at distance $v = 1000$ bp upstream and downstream of $i$, respectively. We use the Wilcoxon rank-sum test to compare the values of $\mathbf{A}[\alpha_i, \beta_i]$ with the values of each of the other two matrices and the highest of the two $p$-values is used. Finally, we correct the $p$-values using the Bonferroni method, and report boundaries with $p < 0.001$.

We also used a minimum local minima depth of 0.01. Depth of local minima (referred to as delta) can be considered similar to "fold change" of any minimum, with respect to the neighboring TAD-score average. These parameters were selected by maximizing the AUC for ROC curves in which the positive set contained CP190 binding sites and the negative set contained inactive regions (black chromatin state from ref. [29]).

In contrast to other published methods to call TADs, this procedure has several advantages: (i) Each boundary is associated to a TAD-separation score and a $p$-value, that are useful to characterize strong vs. weak boundaries, (ii) the TAD score can be easily visualized (e.g., as an genome browser track), which is always useful for visual inspection, and (iii) the computation of boundaries takes only minutes, scaling linearly with the length of the genome. Our method differs from the TopDom method in the following aspects: (i) we compute TAD-separation scores using a z-score matrix while TopDom uses the corrected counts matrix, (ii) we use multiple length ($w$) sizes to compute our TAD-separation score while TopDom uses a single-$w$value, (iii) we compute $p$-values using the 'diamond' $\mathbf{A}[\alpha_l, \beta_l]$ submatrices values in contrast to the 'diamond and triangle' distributions used in TopDom. The triangle distribution contain the intra z-score values between bin $l - w$ and $l$, and the intra z-score values between bin $l$ and $l + w$.

**Validation of boundary quality**. We used the following functional signatures to validate the quality of our boundaries:

*Distance to known insulator co-factor CP190:* since all studied insulators proteins bind to CP190[11,12,14], a sensible quality measure is the overlap of boundaries with CP190 ChIP-seq peaks. For this, we computed the distance of the boundaries to CP190 peaks using bedtools[55] closestBed (Fig. 1b, Supplementary Figure 1h). For comparison, we randomized our boundary positions using bedtools shuffleBed (Supplementary Figure 1h) and estimated the new distances to CP190. ShuffleBed simply assigns a new random position for each boundary anywhere in the genome (excluding heterochromatic and unplaced regions). Finally, we computed the background probability of obtaining the observed overlap between CP190 peaks and Hi-C boundaries using bedtools Fisher's exact test.

*Separation of histone marks:* as boundaries are expected to separate histone marks we used the method described by ref. [5] to quantify the correlation of marks within TADs and between TADs. For this, each TAD was scaled to 15 kb, flanked with a 15 kb region and divided into 1 kb bins. For each bin the mean histone ChIP–chip value was recorded, thus generating a matrix of 2852 TADs (rows) and 45 bins (columns). For this we used computeMatrix from deepTools2. The pairwise pearson correlation value of each column was then computed to produce a matrix of size $45 \times 45$.

**Classification of TADs**. The following histone marks for Kc167 from modEncode[28] were used: H3K36me3, H3K4me3, H3K9me3, and H3K27me3. Other marks that correlate closely to these marks were not included. For example,

H3K9me2 correlates closely with H3K9me3, H3K4me2 with H3K4me3, and so on. The average intensity of the marks over the TAD length was computed using multiBigwigSummary from deepTools2[56]. The resulting matrix was clustered by computing Euclidean distances between the histone marks and applying hierarchical clustering using the complete method. Five clusters were detected (Supplementary Figure 1c) that correspond to the presence of H3K36me3, H3K9me3, H3K4me3, H3K27me3, or none. Analysis of the TADs containing H3K36me3 and H3K4me1 in the genome revealed that that H3K36me3 is present at exons of active genes while H3K4me1 is mostly present at introns and intergenic regions of active genes and less abundant at exons. Thus, noticing that these two marks are complementary for active regions we classified TADs having predominantly these marks as 'active'. For the other clusters we used the same categories as[29]: the cluster of TADs with H3K9me3 was labeled 'HP1' (Heterochromatin Protein 1); the cluster with H3K27me3 was labeled 'repressed' or 'PcG' (Polycomb group) and the cluster with no mark was labeled as inactive.

**Analysis of transcription at boundaries**. In order to analyse transcription at boundaries, we downloaded ribo-depleted RNA-Seq data from modENCODE[57]. All data were mapped to the *Drosophila* (dm3) genome using HISAT2 (v2.0.4)[58] and the reads were summarized per gene using featureCounts (v1.5.0.p1)[59] using options '-p --primary -Q 10'. We used data from Kc167 cells, along with 14 different developmental stages, ranging from embryo to adult. We only used data produced in 2014 in order to avoid batch effects and further confirmed the data quality by clustering the samples by Euclidean distance (Supplementary Figure 2a). We normalized each sample by library size, averaged the counts for replicates, and finally used log transformed counts for all the analysis.

In order to be able to compare all samples from mixed sex, genes on sex chromosomes were excluded for analysis. In total 10,449 autosomal genes were used for expression and variability analysis. Genes were considered expressed if they have a normalized log-count >0. Similar results were obtained using the more stringent cutoff of log counts ≥10, genes with boundary promoters = 45.84% and non-boundary promoter 18.36% ($p$-value = 1.58E − 96, Fisher's exact test). Variability was assessed using coefficient of variation of a gene across all developmental stages along with Kc167 cells. To measure the effect of boundary on nearby gene expression, we first define pairs of adjacent genes in different orientation (convergent, divergent, or tandem) and record the coefficient of variation for each gene in the pair. Then we obtain the scatterplots for all such gene-pairs and compare the results for two different scenarios: (a) where the pairs are separated by a boundary and (b) where there is no such separation. We further tested whether in Kc167 cells, genes within TADs tend to be more correlated in expression amongst them compared to genes lying in the nearby TADs. For this we used a subset of consecutively arranged TADs that have more than one gene inside them. We then used ANOVA to test whether variability gene expression within TADs is different from variability between TADs. As seen in Supplementary Figure 2c, many TADs pairs are significantly different from each other, while very few TADs are significantly different if we randomly assign genes to TADs.

**Identification of boundary motifs**. We took the list of boundaries and expanded them by 500 bp on each side. To avoid false positives, repetitive regions from the sequences of those boundaries were replaced by 'N's and any region with >10% of 'N's was removed. We used MEME-chip[20] to identify enriched DNA motifs; MEME-chip internally computes motifs using two methods, DREME[60] and MEME[27]. DREME aims to quickly identify short motifs while MEME identifies larger overrepresented sequences (at the expense of significantly longer processing times).

To obtain the position-weight matrices of insulator motifs we ran MEME-chip[20] on the peaks called using MACS2[61]. We selected the highest scoring motif for each case which invariably corresponded to the motif reported for the protein. ChIP-seq data sources are found in Supplementary Table 4.

**Enrichment of motifs using control background**. For promoter boundaries a control background composed of all *Drosophila* gene promoters was used to test the enrichment. We downloaded *Drosophila* genes (dm3 assembly) from UCSC table browser[62] and selected the sequences 200 bp upstream and 50 bp downstream of the transcription start site as core promoter sequences.

We classified these promoter sequences as boundary if they were within 500 bp of a boundary, or non-boundary (control background) if they were farther than 2000 bp from a boundary. Repetitive regions from the sequences of those promoters were replaced by 'N's and any region with >10% of 'N's was removed. In total, 10,529 background promoters and 1944 boundary promoters were used.

We used two different methods to assess the enrichment of the de novo and known motifs in boundary promoters with respect the control background, namely Ame[21] from the MEME-suite and a method based on the predicted binding affinity given by TRAP[22] that works as follows: for each motif, the log(TRAP score) distribution was computed for both the background and the boundary promoters. The Wilcoxon rank-sum test was then used to test for differences in the distributions. The $p$-values obtained were corrected using FDR. For Ame, we use total hits as scoring method and Fisher's test for estimating enrichments. We tested

all de novo motifs identified either by MEME or by DREME and all known motifs associated to insulators and CP190 co-factors, as well as all core-promoter motifs.

We also used as control active genes in Kc167. To make this control, we selected those genes that overlapped with the yellow and red chromatin states from ref.[29] that are indicative of active chromatin in *Drosophila* Kc cells. The enrichment results were similar to the ones using a more broader list of genes for background.

For the boundaries that are not at promoters we used non-promoter open chromatin sequences obtained from DNase-seq[28] as control. In this case, we used 1665 background open chromatin regions and 655 non-promoter boundaries.

**Pausing index**. Pausing index for all *D. melanogaster* promoters was computed as the ratio of Pol-II ChIP-seq coverage at promoter over coverage at gene body. We used the ChIP-seq data for RNA Pol-II from Li et al.[15]. The promoter region was defined as in the previous section (200 bp downstream, 50 bp upstream of transcription start site). The gene body was defined as the region between 50 bp downstream of the transcription start site and the gene end. We used the maximum coverage for the promoters and the median coverage for the gene body.

**Processing of ChIP-seq data**. The data sources are listed in Supplementary Table 4. For each ChIP-seq data used, we downloaded the respective fastq files and aligned them in the dm3 assembly using Bowtie2[63]. MACS2 was used to identify peaks for each of the proteins[61]. For the respective data sources we downloaded input sequences and aligned them as the ChIP-seq data. bamCompare and bamCoverage from deepTools2[56] were used to create normalized coverage bigwig files.

MEME-chip[20] was used to identify motifs based on the MACS2 peaks. The resulting motifs can be seen in Supplementary Table 1.

**Clustering of motifs**. We used the promoters (200 bp upstream 50 bp downstream) annotated as boundaries and computed the log(TRAP score) for the Beaf-32 motif, motif-1 (M1BP), motif-6, motif-7 (ZIPIC), and motif-8. The scores for each motif were converted to bigwig files and clustered using hierarchical clustering from deepTools2[56].

All boundaries that were further than 2000 bp of a promoter were centered at the nearest CP190 ChIP-seq peaks within 2000 bp, otherwise the boundary position was not modified. Log(TRAP score) for CTCF, Ibf, and Su(Hw) were computed for these regions and clustered as previously described.

We used hierarchical clustering based on Euclidian distance and the Ward method. The cluster number used was 13 for promoter boundaries and 9 for non-promoter boundaries. In each case, the group composed only of low-TRAP scores was removed. After clustering, the groups were manually ordered to produce the left panel of Fig. 3c. Scale of each heatmap was manually adjusted based on the range of TRAP scores found at the clusters for each motif (Supplementary Figure 3a). The log2 ratio of ChIP-seq/input for the different proteins was used for the center and right panels of Fig. 3c. Each heatmap is centered on the boundary and extended ±5000 bp. Scale of the heatmaps was adjusted based on the log2 ChIP/input for the protein in the respective cluster (Supplementary Figure 3b).

**Motif presence**. In general, sequence motifs occur with different strengths at different genomic loci, and the notion of presence/absence is largely dependent on arbitrary thresholds. Therefore much of our analysis is based on binding scores of motifs rather than their binary presence, which we invoke only for the purpose of visualization (Fig. 5e, Supplementary Figure 3c) and combinatorial motif analysis (Fig. 4). For Fig. 3d and Supplementary Figure 3, we considered a motif as present at a boundary if the TRAP score was equal or higher than the minimum log(TRAP score) identified for the clusters in Fig. 3c (the distribution of the log(TRAP scores) can be seen in Supplementary Figure 3a). The thresholds used were: ZIPIC motif −4.7, Beaf-32 −5, M1BP −4.5, motif-6 −3, motif-8 −2, Ibf −4, CTCF −4, and Su (Hw) −3. For GAF, Pita and Zw5 motifs we used FIMO[64] with the following parameters: '--max-strand --thresh 1e − 3'. For analysis of motif combinations at boundaries, we also require that the motifs are accompanied by the corresponding ChIP-seq peaks. For motif-6 and motif-8 whose binding proteins are not known, we require that the motif is on an accessible region. For this we use the peaks from the DNAse-seq data[28].

**Boundary prediction and feature ranking**. We performed boundary prediction at all *Drosophila* promoters using motif TRAP scores for various transcription factors and DNAse-seq signals as features. We utilized methods ranging from simple to complex (linear models, logistic regression, random forest, and stochastic gradient boosting), with the primary purpose to rank the features by importance in boundary prediction. Pre-filtering was done to remove highly correlated features (pearson $R >$ 40%). Linear model and random forest was performed using the package Caret[65], while logistic regression was performed using package glmnet[66] in R.

Linear model was used with stepwise feature selection algorithm to predict boundary score from features by selecting the combination of features that minimizes the Akaike Information Criteria (AIC). Logistic regression, Random forest, and gradient boosting were used to classify the promoters into boundary and non-boundary, with additional feature selection performed using lasso, for logistic regression. We performed 10-fold cross validation while training all classification models. To evaluate model accuracy, the data were randomly divided

into training (60%) and test (40%) data sets and the sensitivity and specificity was calculated for test predictions. Lasso and gradient boosting models show highly similar sensitivity and specificity when used on an independent test data set, compared to when same data set was used for prediction, suggesting they are robust and less prone to overfitting.

Linear model predicted the boundary scores on the test data set with overall Spearman correlation of 37.6%, while logistic regression and random forest performed predictions with around 73–78% accuracy. After obtaining the best model in each scenario, we ranked the features by their importance in prediction, using the 'varImp' function from Caret. 'varImp' selects a variable importance predictor based on the model type, which is calculated for each parameter in the model (https://topepo.github.io/caret/variable-importance.html). Briefly, the importance score for linear model is the absolute value of the $t$-statistic for the model parameter, for lasso, it is the absolute value of final coefficients, for gbm it is the relative influence score as described in Friedman[67], and for random forest it is the difference between the classification error-rate for the out-of-bag portion of data and a permuted predictor variable, averaged over all trees and normalized by the standard deviation of the differences (https://www.stat.berkeley.edu/~breiman/RandomForests/cc_home.htm#varimp). All importance scores are then scaled between 0 and 100 to compare them together.

**Knockdown of M1BP and Beaf-32.** S2 cells (obtained from Michael Boutros lab) were cultured in Express Five SFM (Thermo Fisher Scientific) supplemented with glutamax, at 27 °C at a density of 1–16 million/ml. All cell lines are regularly checked for the absence of mycoplasma by PCR detection kit (Jena Bioscience PP-401). dsRNA was generated using HiScribe T7 High Yield RNA Synthesis Kit (NEB) and purified with MEGAclear (Ambion). The dsRNA was heated to 65 °C for 5 min and left to cool to room temperature. Primers used for generating M1BP dsRNA are from ref. [17]. Primers used for generating Beaf-32 and GST dsRNABeaf-32_fwd 5′-ttaatacgactcactatagggAATCACGAGGAGCTCACCAA-3′

Beaf-32_rev 5′-ttaatacgactcactatagggCTACTCATCCTTGGCAAGCG-3′
GST_fwd 5′-ttaatacgactcactatagggAGATATCAATTTGTGGGATAGCT-3′
GST_rev 5′-ttaatacgactcactatagggAGATTTTGGATATTAGATACGGT-3′

For knockdown experiments, 7.5 million cells were transfected with 100 µg of dsRNA on 10 cm dishes, using Lipofectamine RNAiMAX Reagent (Thermo Fisher Scientific). The M1BP knockdown lasted for 7 days while that of Beaf-32 lasted for 4 days. For M1BP-Beaf-32 double knockdown, cells were first treated with M1BP dsRNA at day 1, followed by treatment with Beaf-32 dsRNA at day 3, and were collected at day 7. All knockdown experiments were performed in two biological replicates and samples were processed in the same batch.

**Western blot to determine knockdown efficiency.** Protein depletion efficiencies were verified with western blot using specific antibodies against Beaf-32 (1:2000, Creative Diagnostics, CABT-BL422), M1BP (1:1000, gift from David Gilmour, PennState) and Actin (1:2000, Santa Cruz Sc-1616). Blots were incubated overnight at 4 degrees. HRP-coupled goat Anti-Rabbit IgG (1:10,000, GE healthcare) was used as secondary antibody, followed by incubation for 1 h at room temperature. Results were measured using ChemiDoc Imaging system (Bio Rad).

**In situ Hi-C of knockdown and control cells.** We used a modified version of the in situ Hi-C protocol[5] to generate our Hi-C maps. All samples were processed in the same batch. S2 cells were fixed with 2% formaldehyde for 10 min at room temperature. Glycine at 125 mM final concentration was added to the plates followed by 5 min incubation. After two washes in PBS, cells were scraped off the plate and pelleted. Each pellet (10–50 million of cells) was resuspended in 1 ml of lysis buffer (10 mM Tris-HCl, pH 8, 10 mM NaCl, 0.2% IGEPAL CA-630) and nuclei were extracted by sonication following the NEXON protocol[68] (Covaris E220 sonicator, settings: 75 W peak power, 2% duty factor, 200 cycles/burst, for 60–90 s until about 70% of intact nuclei were released). Nuclei were pelleted and resuspended in 0.5% SDS to permeabilize the nuclear membrane and to make chromatin accessible for restriction digestion. After 10 min incubation at room temperature, SDS was quenched adding 1% Triton X-100 (final concentration) and 1X of NEBuffer 3.1 (NEB, B7203S). Nuclei were digested overnight at 37 °C on a rocking platform using DpnII (NEB, R0543M, ~1–5 units per million cells). Prior and after digestion, an aliquot of nuclei was set aside for digestion quality control and DNA quantification. Biotin incorporation was carried out in a final volume of 150 µl using these reaction conditions: 50 mM of each dNTPs, replacing dCTP with biotin-14-dCTP (Life Technologies, 19518-018), 1 U of Klenow (NEB, M0210L) per microgram of DNA, at 25 °C for 1 h to promote fill-in. Ligase mix was added (1X Ligation buffer NEB B0202, 0.8% Triton X-100, 0.1 mg/ml BSA, 2000 U T4 DNA ligase NEB M0202S, final sample volume 1.2 ml) and samples were incubated for 4 h at room temperature under rotation. Nuclei were pelleted, resuspended in SDS buffer (10 mM Tris-HCl, pH 8, 1 mM EDTA, 1% SDS), lysed, de-crosslinked and de-proteinized overnight at 68 °C. DNA was precipitated and sonicated to the size range of 100–600 bp. Biotinylated Hi-C DNA in 1X binding buffer (5 mM Tris-HCl, pH 8, 0.5 mM EDTA, 1 M NaCl) was pulled down using Dynabeads MyOne Streptavidin C1 (Life Technologies, 650.01), using 5 µl of beads per microgram of DNA, pre-washed in 1X binding buffer. Beads were washed twice in Tween wash buffer (5 mM Tris-HCl, pH 8, 0.5 mM EDTA, 1 M NaCl, 0.05% Tween-20) and twice in EB (10 mM Tris-HCl, pH 8). A small aliquot of beads-

bound DNA was kept to measure the DNA concentration, eluting the DNA by incubation at 98 °C for 10 min. Quantity of 10–50 ng of DNA bound to beads was used for library preparation using a modification of the NEBNext Ultra II DNA library preparation workflow (NEB, E7645). DNA bound to beads was then end-repaired, A-tailed, adaptor-ligated using manufacturer's instruction. Beads were reclaimed on a magnet, washed once in EB and eluted at 98 °C for 10 min. DNA was USER-treated to open the adapters in meantime to PCR amplification. Libraries were sequenced paired-end, with a read length of 75 bp, on Illumina HiSeq 3000.

**Cell cycle analysis using FACS.** Formaldehyde-fixed cells (2% for 10 min) were resuspended in 500 µl of permeabilization buffer (1% BSA, 0.5% Triton X-100 in PBS) and incubated for 15 min at room temperature under mixing. Cells were then incubated overnight at 4 °C with a 1:500 dilution of anti-H3 Ser10 phosphorylated antibody (Abcam, ab5176) and washed twice with permeabilization buffer. Cells were pelleted and resuspended in 400 µl of permeabilization buffer. One microliter of secondary antibody (anti-rabbit conjugated with Alexa 633 fluorophore, A21070, Life Technologies) was added and cells were incubated for 2 h at room temperature (protected from light). After two washes using the permeabilization solution, cells were resuspended in 500 µl of PBS and treated with 5 µl of RNase A for 30 min at room temperature. Nuclei were stained using 1 µg/ml of DAPI prior sorting. Samples were analyzed by flow cytometry using BD LSRFortessa instrument. The data have been analyzed using FACSDiva.

**Hi-C processing of Beaf-32 and M1BP knockdowns.** Hi-C samples for GST control and knockdowns were processed as all other Hi-C samples using HiCExplorer. The bins are based on the restriction fragments for DpnII. The statistics of data processing are shown in Supplementary Figure 6d-f and Supplementary Table 5.

**Code availability.** HiCExplorer code is available online at: https://github.com/deeptools/HiCExplorer/. HiCBrowser code is available at: https://github.com/deeptools/HiCBrowser

**Data availability.** The processed data from ChIP-Seq and Hi-C samples obtained from online sources can be found at: http://chorogenome.ie-freiburg.mpg.de/data_sources.html. Sequencing data for the in situ Hi-C experiments described in this article are available at NCBI GEO under accession: GSE97965.

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

## Acknowledgements

We would like to thank David Gilmour (PennState University) for providing the M1BP antibody. Also, we would like to thank Ulrike Bönisch and her team at the Deep-Sequencing Facility and Andreas Würch from the FACS facility of the MPI-IE. We further thank Wilhelm Rüsing for IT support, Diana Santacruz for critical reading of the manuscript and Gina Renschler for the help in interpreting FACS results. We also thank Victor Corces Lab whose published data have been instrumental in this research. This work was supported by the German Research Foundation [SFB 992 "Medical Epigenetics"] awarded to A.A. and T.M. and a grant from the Federal Ministry of Education and Research through the German Epigenome Programme DEEP [01KU1216G] awarded to T.M.

## Author contributions

F.R. developed HiCExplorer, HiCBrowser and the Chorogenome navigator with support from V.B., B A.G. and J.V. F.R. and V.B. performed all the analysis and designed the experiments, with inputs from T.M. L.A. and K.C.L. performed the experiments. T.M. supervised all aspects of the analysis, while B.H. and A.A. supervised J.V. and K.C.L., respectively. F.R. and V.B. wrote the manuscript with input from all other authors.

## Additional information

**Competing interests:** The authors declare no competing financial interest.

