## [Peer Review File · Nature Communications]

Reviewers' comments:

Reviewer #1 (Remarks to the Author):

Reading the authors' rebuttal and the revised version of this manuscript, it seems like the authors are trying to argue about minimal novelty points to differentiate their study from previous ones. Instead, it might be more productive if the authors would take the reviewers' comments on board –which have been so far instrumental in pointing out major flaws in the conclusions reached in previous submissions– and would use them to produce a comprehensive and up-to-date picture of 3D chromatin organisation in *Drosophila*.

In my opinion, the following points still need to be addressed before the manuscript could be considered ready for publication.

1. Analysis of constitutive gene expression

As highlighted in my previous reviews, the link with constitutive expression in flies has been made before directly or indirectly through analysis of chromatin signatures by Ulianov et al., (Genome Research 2016); Cubenas-Potts et al., (NAR 2016); Hug et al., (Cell 2017); El-Sharnouby et al., (PLoS ONE 2017); and more recently Schauer et al., (EMBO Reports 2017). Here, the authors use higher resolution TAD calls and a slightly different measurement of gene expression by looking at transcriptional variability to arrive at the same conclusions. This is just alternative confirmatory evidence of what was known before and, as such, it should be stated in the manuscript and properly discussed (please note that the authors only refer to Hou et al., 2012 in their discussion of these findings and appear to ignore any of the other relevant literature suggested before).

2. I still do not understand the following sentence despite the edits:

[line 154] "We found that 95.6% of genes which have a TAD boundary at their promoter are expressed in Kc167 cells compared to only 75.3% of genes which do not have a boundary at their promoter (p-value = 2.19E-80, Fisher's exact test)."

This would mean that essentially all genes with promoters at boundaries are expressed (which is consistent with being constitutively expressed genes), as well as >75% of genes without promoters at TADs. But this would leave only the <25% of genes without promoters at TADs silent in the genome, resulting in what seems like a really high proportion of the genome being expressed in Kc167 cells.

Could the authors please state the total number of genes in each category so it can be clarified whether "expression" is being called with a too relaxed threshold (a normalised log-count of 1 or more [line 600]) or whether this only includes a subset of genes in the genome? Is there alternative evidence for some many genes being expressed in Kc167 cells? One could test this by for example looking at Ser2 RNA PolII binding, GRO-seq, NET-seq, or equivalent datasets to support this claim.

Similarly as suggested by reviewer #3 before in a different context, I'd suggest removing "only" from the statement, since this is not an objective qualifier.

3. Motif analysis

As mentioned by reviewer #2, many of the motifs found in this manuscript have been reported by a previous publication from the Vaquerizas laboratory (Hug et al., Cell 2017) looking at the emergence of TADs during early embryonic development, and this should be properly reflected and discussed in a revised version of the manuscript. The authors reach similar conclusions since many TAD boundaries seem to be invariant once established, which helps in cementing the role of these motifs in chromatin organisation.

Within the context of motif analysis, the authors should also include in their discussion previous relevant work by the Akhtar and Becker laboratories (Lam et al., PLoS Genetics 2012; and Feller et al., NAR 2012, respectively) that have already linked the presence of some of these motifs to the regulation of constitutively expressed genes in *Drosophila*, which would nicely complement the analyses of constitutive gene expression presented in the manuscript.

4. Changes in chromatin organisation upon depletion of Beaf-32 and M1BP

The authors now confirm that their previous conclusions for M1BP depleted cells are most likely artefactual, since these cells are arrested in M phase. This makes it difficult to determine the direct effect of the loss of M1BP on chromatin organisation, if any, which the authors now acknowledge at the end of the discussion.

The authors argue that their experiments are working properly based on their reproducibility and that their data are of good quality. However, similarly as highlighted by reviewer #2, I still find the data presented in Fig 6B not convincing. In the revision, the authors compare their M phase-arrested cells data with those of mitotic embryos (Fig 6E), but these fail to show long-range contacts, which have been reported before in mitotic fly embryos (Hug et al., Cell 2017) as well as in mammalian systems (Naumova et al., Science 2013; Nagano et al., Nature 2017) and are hallmarks of mitotic chromosomes. This suggests that there might be further potential technical artefacts in their data. Instead, based on the comparison of the total number of inter-chromosomal interactions, the authors hypothesise a chromosome intermingling phenotype in these cells [line 327], but this is not tested further.

This would certainly be a puzzling phenotype, assuming that the experiments worked properly (this is also complicated to figure out since, as I highlighted in my previous review, the cis/trans ratio of interactions is usually indicative of the quality of a Hi-C dataset), since I find it very complicated to understand how it would be possible that there are no short-range interactions and no long-range interactions at the same time. Do the authors have any explanation for this? Maybe the authors could include immunohistochemistry or DAPI stainings for these cells to at least evaluate the overall organisation of chromatin in those M phase arrested cells, which might help in explaining the proposed phenotype.

Within this context, and because of the inability to determine direct effects, I would suggest that the authors edit the following sentence in the abstract, since the sentence can be understood as a causality statement as if the depletion of M1BP would be responsible for changes in chromatin conformation that would then lead to cell cycle arrest, which at the moment the authors are unable to show:

[line 42] "The depletion of Beaf-32 has no effect on chromosome organisation while depletion of M1BP show a dramatic effect concurrent with a cell cycle arrest."

The following sentences should also be edited, similarly as above, if the authors are not able to find a convincing explanation for the M1BP Hi-C datasets (in which case maybe the authors should stress the M phase arrest phenotype of the M1BP knockdown samples and move the corresponding Hi-C data to the supplementary material):

[line 97] "We show that depletion of Beaf-32 has no major effect on chromosome organisation, while the depletion of M1BP leads an increase in inter- chromosomal contacts along with an arrest in cell cycle."

[line 362] "Using in-situ Hi-C upon M1BP depletion we observe large changes in the Drosophila genome organisation."

Reviewer #2 (Remarks to the Author):

This is an interesting paper, and definately at the level for Nature Communications. I would accept it as is.

Reviewer #3 (Remarks to the Author):

I have now re-read the manuscript by Denke colleagues and consider the it well written and the conclusion interesting. However, I also concur with other reviewers that the new Hi-C data (although reproducible) has some quality metrics that would need to be re-checked of justified by the effect of the mutants.

To assess the quality of the Hi-C data, I have mapped and cleaned the Hi-C experiments using an alternative software package (thus the results should not be 100% identical of those presented in the manuscript). I have run one replica for the wild-type experiments and two replicates for the mutant conditions. Unfortunately, there are clear signs that some problems may have been present for the mutant experiments. At this point, it is difficult for me to assess whether the results are consequence of low quality in the data or the actual effect of the experiment. For example, the percentage of cis interaction is low compared to standard Hi-C experiments (including the wild-type conditions by the same authors). This is confirmed by the slope of the decay of interactions, it's -1 for wild-type conditions (as expected) and -0.5 for mutant conditions (in both replicates).

Once the data is mapped, then one has to filter-out non-valid interactions. The filtering clearly shows that wild-type conditions are good, while mutant experiments show all sort of problems with large proportion of dangling ends (bad ligation), and high random breaks of the DNA (which is in this case more problematic). In fact, it is likely that the huge number of random breaks results in an affected slope in those experiment as well as the affected percentage of cis/trans interactions.

Said that, the valid pairs for each of the mutant experiments (~20%, which is very low for standard conditions of Hi-C experiments) could be correct but one cannot be 100% sure these experiments are clean. I am attaching to this review an Excel file with the data for each of the mappings for replicates 1 and 2 for the experiments.

I do not have at this point any other particular concern for the rest of the manuscript. However, the authors should make a case for the mutant Hi-C datasets to be accepted given the worry about the quality of the experiments.

Reviewers' comments:

Reviewer #1 (Remarks to the Author):

Reading the authors' rebuttal and the revised version of this manuscript, it seems like the authors are trying to argue about minimal novelty points to differentiate their study from previous ones. Instead, it might be more productive if the authors would take the reviewers' comments on board –which have been so far instrumental in pointing out major flaws in the conclusions reached in previous submissions– and would use them to produce a comprehensive and up-to-date picture of 3D chromatin organisation in *Drosophila*.

In my opinion, the following points still need to be addressed before the manuscript could be considered ready for publication.

1. Analysis of constitutive gene expression

As highlighted in my previous reviews, the link with constitutive expression in flies has been made before directly or indirectly through analysis of chromatin signatures by Ulianov et al., (Genome Research 2016); Cubenas-Potts et al., (NAR 2016); Hug et al., (Cell 2017); El-Sharnouby et al., (PLoS ONE 2017); and more recently Schauer et al., (EMBO Reports 2017). Here, the authors use higher resolution TAD calls and a slightly different measurement of gene expression by looking at transcriptional variability to arrive at the same conclusions. This is just alternative confirmatory evidence of what was known before and, as such, it should be stated in the manuscript and properly discussed (please note that the authors only refer to Hou et al., 2012 in their discussion of these findings and appear to ignore any of the other relevant literature suggested before).

We have cited earlier studies that directly link constitutive gene expression to boundaries. The changes are highlighted in the text. Nevertheless, we have already pointed out that earlier research provided contradicting information (as discussed in our manuscript), and our analysis, based on the most accurate boundaries, aims to clarify prior conclusions.

2. I still do not understand the following sentence despite the edits:

[line 154] “We found that 95.6% of genes which have a TAD boundary at their promoter are expressed in Kc167 cells compared to only 75.3% of genes which do not have a boundary at their promoter (p-value = 2.19E-80, Fisher's exact test).”

This would mean that essentially all genes with promoters at boundaries are expressed (which is consistent with being constitutively expressed genes), as well as >75% of genes without promoters at TADs. But this would leave only the <25% of genes without promoters at TADs silent in the genome, resulting in what seems like a really high proportion of the genome being expressed in Kc167 cells.

Could the authors please state the total number of genes in each category so it can be clarified whether “expression” is being called with a too relaxed threshold (a normalised log-count of 1 or more [line 600]) or whether this only includes a subset of genes in the genome? Is there alternative evidence for some many genes being expressed in Kc167 cells? One could test this by for example looking at Ser2 RNA PolIII binding, GRO-seq, NET-seq, or equivalent datasets to support this claim.

We have clarified the number of genes and cutoff used to calculate gene expression threshold. This is highlighted in the results section: "Specific gene orientation and transcription marks boundaries" on page 5. Additionally, we mention that increasing the expression cutoff will result in a smaller number of expressed genes, but does not affect our conclusion that boundaries tend to have higher expression. This is highlighted in the methods section "Analysis of transcription at boundaries", page 18.

Similarly as suggested by reviewer #3 before in a different context, I'd suggest removing “only” from the statement, since this is not an objective qualifier.

We have removed “only” from the statement as suggested. Now the sentence (highlighted text in page 5) reads: "We found that 95.6% of genes which have a TAD boundary at their promoter are expressed in Kc167 cells (1,244 out of 1,300) compared to 75.3% of genes which do not have a boundary at their promoter (6,892 out of 9,149, p-value = 2.19E-80, Fisher's exact test)."

3. Motif analysis

As mentioned by reviewer #2, many of the motifs found in this manuscript have been reported by a previous publication from the Vaquerizas laboratory (Hug et al., Cell 2017) looking at the emergence of TADs during early embryonic development, and this should be properly reflected and discussed in a revised version of the manuscript. The authors reach similar conclusions since many TAD boundaries seem to be invariant once established, which helps in cementing the role of these motifs in chromatin organisation.

Within the context of motif analysis, the authors should also include in their discussion previous relevant work by the Akhtar and Becker laboratories (Lam et al., PLoS Genetics 2012; and Feller et al., NAR 2012, respectively) that have already linked the presence of some of these motifs to the regulation of constitutively expressed genes in Drosophila, which would nicely complement the analyses of constitutive gene expression presented in the manuscript.

We have now clearly stated and discussed earlier studies that also detect motifs at boundaries, in introduction, results section "Beaf-32 depletion does not affect TAD boundaries" and discussion sections of the manuscript (see highlighted text). We also added the two other citations from related work to our discussion, as suggested by the reviewer (also highlighted in the text, page 11).

4. Changes in chromatin organisation upon depletion of Beaf-32 and M1BP

The authors now confirm that their previous conclusions for M1BP depleted cells are most likely artefactual, since these cells are arrested in M phase. This makes it difficult to determine the direct effect of the loss of M1BP on chromatin organisation, if any, which the authors now acknowledge at the end of the discussion.

The authors argue that their experiments are working properly based on their reproducibility and that their data are of good quality. However, similarly as highlighted by reviewer #2, I still find the data presented in Fig 6B not convincing. In the revision, the authors compare their M phase-arrested cells data with those of mitotic embryos (Fig 6E), but these fail to show long-range contacts, which have been reported before in mitotic fly embryos (Hug et al., Cell 2017) as well as in mammalian systems (Naumova et al., Science 2013; Nagano et al., Nature 2017) and are hallmarks of mitotic chromosomes. This suggests that there might be further potential technical artefacts in their data. Instead, based on the comparison of the total number of inter-chromosomal interactions, the authors hypothesise a chromosome intermingling phenotype in these cells [line 327], but this is not tested further.

This would certainly be a puzzling phenotype, assuming that the experiments worked properly (this is also complicated to figure out since, as I highlighted in my previous review, the cis/trans ratio of interactions is usually indicative of the quality of a Hi-C dataset), since I find it very complicated to understand how it would be possible that there are no short-range interactions and no long-range interactions at the same time. Do the authors have any explanation for this? Maybe the authors could include immunohistochemistry or DAPI stainings for these cells to at least evaluate the overall organisation of chromatin in those M phase arrested cells, which might help in explaining the proposed phenotype.

We agree with the reviewer that the cell cycle arrest makes it difficult to claim that the changes in chromatin organisation are a direct effect of M1BP knockdown. Therefore we have modified the introduction, results section "Beaf-32 depletion does not affect TAD boundaries" and discussion (see highlighted text). These changes do not have any effect on the main message of the paper which centers on the role of DNA motifs in the 3D conformation of flies.

Even though we agree with the comment regarding the direct role of M1BP on boundaries, we disagree that the Hi-C data is of low quality (see also response to reviewer 3). In contrast to mitotic cells from Hug et. al. we see that M1BP knockdowns have a different chromatin organization (Fig. 6C-D). Please note that Hug et. al data represents nuclei from healthy embryo collected at the stage of mitosis, not mitotically arrested cells. We see an increase in inter-chromosomal interactions, which in healthy cell lines could be a concern about Hi-C quality (see response to Reviewer 3). Instead, given the poor state in which M1BP depleted cells grow and the high quality of wild type and Beaf-32 Hi-C processed identically, we argue that the inter-chromosomal interactions in M1BP KD reflect an abnormal chromatin condition. Further experiments can be done to explore this but are beyond the scope of our research. Therefore we have refrained from giving relevance of this issue. Instead, our goal is to raise the awareness to other researchers of the problems of doing Hi-C on knockdowns of proteins that control many housekeeping genes.

Within this context, and because of the inability to determine direct effects, I would suggest that the authors edit the following sentence in the abstract, since the sentence can be understood as a causality statement as if the depletion of M1BP would be responsible for changes in chromatin conformation that would then lead to cell cycle arrest, which at the moment the authors are unable to show:

[line 42] “The depletion of Beaf-32 has no effect on chromosome organisation while depletion of M1BP show a dramatic effect concurrent with a cell cycle arrest.”

The following sentences should also be edited, similarly as above, if the authors are not able to find a convincing explanation for the M1BP Hi-C datasets (in which case maybe the authors should stress the M phase arrest phenotype of the M1BP knockdown samples and move the corresponding Hi-C data to the supplementary material):

[line 97] “We show that depletion of Beaf-32 has no major effect on chromosome organisation, while the depletion of M1BP leads an increase in inter- chromosomal contacts along with an arrest in cell cycle.”

[line 362] “Using in-situ Hi-C upon M1BP depletion we observe large changes in the Drosophila genome organisation.”

We edited all these statements as suggested by the reviewer.

Reviewer #2 (Remarks to the Author):

This is an interesting paper, and definitely at the level for Nature Communications. I would accept it as is.

Reviewer #3 (Remarks to the Author):

I have now re-read the manuscript by Denke colleagues and consider the it well written and the conclusion interesting. However, I also concur with other reviewers that the new Hi-C data (although reproducible) has some quality metrics that would need to be re-checked of justified by the effect of the mutants.

To assess the quality of the Hi-C data, I have mapped and cleaned the Hi-C experiments using an alternative software package (thus the results should not be 100% identical of those presented in the manuscript). I have run one replica for the wild-type experiments and two replicates for the mutant conditions. Unfortunately, there are clear signs that some problems may have been present for the mutant experiments. At this point, it is difficult for me to assess whether the results are consequence of low quality in the data or the actual effect of the experiment. For example, the percentage of cis interaction is low compared to standard Hi-C experiments (including the wild-type conditions by the same authors). This is

confirmed by the slope of the decay of interactions, it's -1 for wild-type conditions (as expected) and -0.5 for mutant conditions (in both replicates).

Once the data is mapped, then one has to filter-out non-valid interactions. The filtering clearly shows that wild-type conditions are good, while mutant experiments show all sort of problems with large proportion of dangling ends (bad ligation), and high random breaks of the DNA (which is in this case more problematic). In fact, it is likely that the huge number of random breaks results in an affected slope in those experiment as well as the affected percentage of cis/trans interactions.

Said that, the valid pairs for each of the mutant experiments (~20%, which is very low for standard conditions of Hi-C experiments) could be correct but one cannot be 100% sure these experiments are clean. I am attaching to this review an Excel file with the data for each of the mappings for replicates 1 and 2 for the experiments.

I do not have at this point any other particular concern for the rest of the manuscript. However, the authors should make a case for the mutant Hi-C datasets to be accepted given the worry about the quality of the experiments.

We thank the reviewer for taking the effort to run quality control metrics on our data. The following is the summary of our response:

- The QC metrics for our samples are good and completely comparable to other works and to the reviewers' own assessment.
- We were concerned that the reviewer reports different values for dangling ends and investigated this extensively (see reviewers figures) to verify the correctness of our measurements.
- Inter-chromosomal contacts indirectly measure random pairs and can not distinguish true biological phenomena from poor Hi-C quality.

The reviewer's QC measurements and HiCExplorer measurements mostly agree. Based on the reviewers values, on average 21% of our Hi-C pairs are valid. In our own assessments this number is 18% (due to a more stringent filtering of reads on our side). This number is comparable to the number of valid pairs for most of the data from (Cubéñas-Potts et al. 2016), re-analyzed in the manuscript, which is 21%. The most efficient Hi-C libraries have around 40% valid reads pairs as the one from (Giorgetti et al. 2016) (See reviewers Fig. 1).

The number of **valid read pairs** represent the overall efficiency of the protocol (digestion by restriction enzyme, re-ligation, pulldown of biotinylated pairs, contamination and PCR duplications) which vary depending on the cell type and condition. However, this number does not reflect the quality of the Hi-C data and does not affect subsequent analysis. Dilution Hi-C protocols produce even lower fraction of valid read pairs (e.g. 12% for Sexton et. al.).

To understand possible causes for invalid read pairs, HiCExplorer QC measures report the number of dangling ends, duplicated pairs, self-circles, one fragment not close to restriction site and same fragment pairs. The number of **dangling ends** is indicative of the re-ligation efficiency. In our case we used biotinylated C which has a reduced re-ligation efficiency

compared to biotinylated A. That is why our samples had increased number of dangling ends compared to other samples (see reviewers Fig. 1).

To estimate the efficiency of the pulldown of biotinylated DNA we report the number of **same fragment pairs**. These are paired reads that are less than 800 bp apart and that do not have any restriction enzyme site in between. They are normal DNA paired end reads, which originated from fragments attached unspecifically to streptavidin beads. In our samples we can see that the number of same fragment pairs (31%) is similar to that of (Giorgetti et al. 2016) (25%), but compared to the data from (Cubeñas-Potts et al. 2016) (8%) there is room for improvement. Indeed, after adding more stringent washing we have decreased the number of 'same fragment pairs' to 10-15% in recent in-house Hi-C experiments.

For our Beaf-32 knockdown sample (replicate 1), the reviewer assessment reports 47% of dangling ends while we find 11%. We investigated this further and could not find evidence for a large amount of dangling ends. Reviewers Figure 2 shows unfiltered Hi-C reads (top) and filtered Hi-C reads (bottom) from the Beaf-32 KD r1 sample. The top panel contains a small fraction of dangling ends. They are identified because the start of the read matches the overhang of the restriction sequence (which is different for every restriction enzyme). We think that in the reviewer's measurement, the **dangling ends** and the '**same fragment**' measurements are merged. For the Beaf-32 r1 KD sample we measure 11% dangling ends and 41% same fragment pairs which is closer to the reviewer's number of 47%. In general, for all eight samples, we could see that by adding 'dangling ends' and 'same fragment' pairs we reach similar numbers than those from the reviewer. We do not recommend merging these two measurements because, as explained earlier, each value helps to determine which part of the protocol may be improved to reduce the cost of sequencing non Hi-C data.

The reviewer's assessment reports **random breaks** which, following the definition by (Servant et al. 2015), are caused by fragments outside the expected range and could be indicative of star effect of the restriction enzyme. HiCExplorer reports those pairs as '**One mate not close to restriction site**' (Reviewers Fig. 1). However, given the amount and distribution of pairs labeled as **random breaks** by the reviewer's assessment we assume that this is rather measuring the **inter-chromosomal** contacts.

As shown by (Kalhor et al. 2011) an important measurement to identify the quality of the Hi-C data is the **inter-chromosomal contacts** which are expected to be below 20% of total valid read pairs in healthy cells. This measurement is an *indirect estimation of random ligations* (Hi-C ligations that occur for reasons other than spatial proximity) in the Hi-C library preparation. The assumption is that chromosomes occupy their own territories and therefore most contacts occur within chromosomes. Although this is true for healthy cells, it is unknown how the inter-chromosomal contacts are affected in mitotically arrested cells with defects in chromatin architecture. In our opinion the best solution is to prepare Hi-C libraries from healthy and mutant cells at the same time such that the healthy Hi-C libraries could be used to inform us about the expected levels of random ligations (as we have done in this study).

Compared to the original dilution Hi-C protocol (Lieberman-Aiden et al. 2009), the in-situ Hi-C (Rao et al. 2014) and TCC (Kalhor et al. 2011) protocols reduce the possibility of forming random ligations in the solution significantly. Our experiments, based on the In-situ Hi-C protocol, achieve an average of 8% inter-chromosomal interactions for healthy cells (wild type and Beaf-32 KD) which indicates excellent Hi-C data quality. In contrast, the data for (Cubéñas-Potts et al. 2016) has 14% inter-chromosomal interactions and the dilution Hi-C data from (Giorgetti et al. 2016) has 39% inter-chromosomal contacts (Reviewers Fig. 1).

Our four M1BP knockdown samples have inter-chromosomal contacts of around 45%. Yet, these samples were processed in identical way, by the same person, at the same time as the other four samples that have ~8% of inter-chromosomal contacts. Thus, in contrast to the reviewers opinion, we argue that these libraries do not suffer from a technical artifact and that the high inter-chromosomal contacts are reflecting the biological effect on the chromatin. M1BP knockdown cells appear unhealthy, grow slower in culture and get arrested in M-Phase, as shown in the manuscript. Given that M1BP is a transcription factor of constitutively expressed genes (Li and Gilmour 2013) we could expect significant problems in the knockdown cells. Although we do not have other experiments to demonstrate chromatin effects in M1BP depleted cells we think that we have reasonable evidence to trust the quality of our Hi-C libraries. Further experiments to validate chromatin problems in M1BP knockdown cells, although desirable, are beyond the scope of the manuscript.

Thus, based on current observations and our experience in processing many Hi-C samples, we stand behind the quality of these Hi-C data and the relevance and correctness of our QC measurements. Because of this, we would like to keep the data as part of the publication as this will serve as a guide to other researchers planning similar experiments.

A summary of this response about Hi-C data quality was added to the methods section of the manuscript (highlighted text on page 26, under heading 'Hi-C processing of Beaf-32 and M1BP knockdowns')

As stated earlier, we indicate that the results are inconclusive about the role of M1BP on boundaries and therefore we have moved these results into supplementary figures. This does not affect the conclusions of the manuscript and is discussed appropriately in the manuscript. The relevant changes in abstract, introduction, results (section "Beaf-32 depletion does not affect TAD boundaries") and discussion are highlighted.

References

- Cubéñas-Potts, Caelin, M. Jordan Rowley, Xiaowen Lyu, Ge Li, Elissa P. Lei, and Victor G. Corces. 2016. "Different Enhancer Classes in Drosophila Bind Distinct Architectural Proteins and Mediate Unique Chromatin Interactions and 3D Architecture." *Nucleic Acids Research*, November. doi:10.1093/nar/gkw1114.
- Giorgetti, Luca, Bryan R. Lajoie, Ava C. Carter, Mikael Attia, Ye Zhan, Jin Xu, Chong Jian Chen, et al. 2016. "Structural Organization of the Inactive X Chromosome in the Mouse." *Nature* 535 (7613): 575–79.
- Kalhor, Reza, Harianto Tjong, Nimanthi Jayathilaka, Frank Alber, and Lin Chen. 2011.

“Genome Architectures Revealed by Tethered Chromosome Conformation Capture and Population-Based Modeling.” *Nature Biotechnology* 30 (1). Nature Publishing Group: 90–98.

Lieberman-Aiden, Erez, Nynke L. van Berkum, Louise Williams, Maxim Imakaev, Tobias Ragozy, Agnes Telling, Ido Amit, et al. 2009. “Comprehensive Mapping of Long-Range Interactions Reveals Folding Principles of the Human Genome.” *Science* 326 (5950): 289–93.

Li, Jian, and David S. Gilmour. 2013. “Distinct Mechanisms of Transcriptional Pausing Orchestrated by GAGA Factor and M1BP, a Novel Transcription Factor.” *The EMBO Journal* 32 (13): 1829–41.

Rao, Suhas S. P., Miriam H. Huntley, Neva C. Durand, Elena K. Stamenova, Ivan D. Bochkov, James T. Robinson, Adrian L. Sanborn, et al. 2014. “A 3D Map of the Human Genome at Kilobase Resolution Reveals Principles of Chromatin Looping.” *Cell* 159 (7). Elsevier Inc.: 1665–80.

Servant, Nicolas, Nelle Varoquaux, Bryan R. Lajoie, Eric Viara, Chong-Jian Chen, Jean-Philippe Vert, Edith Heard, Job Dekker, and Emmanuel Barillot. 2015. “HiC-Pro: An Optimized and Flexible Pipeline for Hi-C Data Processing.” *Genome Biology* 16 (December): 259.

Reviewers Figure 1

Reviewer QC of our data

Own data

Li et al. data

Giorgetti et al. data

Figure 1. Comparison of QC measurements from reviewer #3 and from HiCExplorer.

Apart from our own 8 samples, we included the QC for the data used in the manuscript from (Cubéñas-Potts et al. 2016) and the recently published data from (Giorgetti et al. 2016) in mouse cells. HiCExplorer provides QC data for the number of mapped reads and multimapping reads (left panels); for ‘One mate not close to restriction site’, dangling ends, duplications, ‘same fragment’, ‘self-circles’ and ‘self ligation’ (middle panel) that is useful to determine which steps of the Hi-C protocol may be affecting the number of valid Hi-C pairs but is not related to the quality of the valid Hi-C pairs. From the right panel, the number of inter-chromosomal interactions is normally used as an indirect estimation of ‘random ligations’ to assess the quality of the Hi-C data.

Reviewers Figure 2

Unfiltered reads

Filtered reads

Figure 2. Dangling ends. Dangling ends are DNA fragments that did not re-ligate after digestion with the restriction enzyme. They are easy to identify because they start with the restriction enzyme overhang sequence. In a genome browser they appear as reads whose start is exactly at the restriction site. Red boxes indicate dangling ends present in the unfiltered reads image. Notice that many reads **end** at the restriction site and those are valid Hi-C reads. For this example, the total number of dangling ends with respect to total mapped and unique reads was 11%. In the filtered reads image, the dangling ends are removed by HiCExplorer as well as 'same fragment' and duplicated reads among others.

REVIEWERS' COMMENTS:

Reviewer #1 (Remarks to the Author):

The authors have successfully addressed my comments and I support publication of this manuscript.

Reviewer #3 (Remarks to the Author):

I appreciate the effort that the authors have made in clarifying to the readers the quality of their experiments. In fact, I would even encourage them to further discuss them in the manuscript and not only have it in the Methods section.